# Programming with Pixels: Can Computer-Use Agents do Software Engineering?

**Pranjal Aggarwal & Sean Welleck**
Carnegie Mellon University
{`pranjala, swelleck`}`@cs.cmu.edu`

## Abstract

Computer-use agents (CUAs) hold the promise of performing a wide variety of general tasks, but current evaluations have primarily focused on simple scenarios. It therefore remains unclear whether such generalist agents can automate more sophisticated and specialized work such as software engineering (SWE). To investigate this, we introduce `Programming with Pixels` (PwP), the first comprehensive computer-use environment for software engineering, where agents visually control an IDE to perform diverse software engineering tasks. To enable holistic evaluation, we also introduce `PwP-Bench`, a benchmark of 15 existing and new software-engineering tasks spanning multiple modalities, programming languages, and skillsets. We perform an extensive evaluation of state-of-the-art open-weight and closed-weight CUAs and find that when interacting purely visually, they perform significantly worse than specialized coding agents. However, when the same CUAs are given direct access to just two APIs—file editing and bash operations—performance jumps, often reaching the levels of specialized agents despite having a task-agnostic design. Furthermore, when given access to additional IDE tools via text APIs, all models show further gains. Our analysis shows that current CUAs fall short mainly due to limited visual grounding and the inability to take full advantage of the rich environment, leaving clear room for future improvements. `PwP` establishes software engineering as a natural domain for benchmarking whether generalist computer-use agents can reach specialist-level performance on sophisticated tasks. [1]

## 1 Introduction

Computer-use agents (CUAs) hold the promise of automating a wide range of economically valuable tasks by acting through primitive actions such as clicking, typing, and observing digital screens, potentially obviating the need for specialized AI agent action interfaces (Anthropic, 2024; OpenAI, 2025; Yang et al., 2024a). However, current evaluations have primarily focused on simple tasks such as web navigation (Koh et al., 2024), basic document editing, or tweaking settings in operating systems (Xie et al., 2024; Bonatti et al., 2024). Therefore, it remains unclear whether current generalist computer-use agents can automate more sophisticated and specialized tasks such as software engineering. In this work, we specifically study how well the current generation of computer-use agents can do software engineering and identify their key limitations.

The choice of using software engineering as the test domain is motivated by two primary reasons. First, software engineering represents an economically important and practically challenging task. Second, the field of AI software-engineering agents (SWE agents) has produced numerous specialized agents that use hand-engineered APIs for specific operations (Yang et al., 2024a; Wang et al., 2024b; Xia et al., 2024), providing strong baselines for comparison. These agents use custom functions such as file editing, code search, and repository management, with each tool requiring significant engineering effort and domain expertise. For instance, SWE-agent (Yang et al., 2024a) uses language-specific parsers and editing commands, while Agentless (Xia et al., 2024) relies on Python-specific abstract syntax trees. This specialization has yielded strong performance, but it

---

[1]Code and data released at `https://programmingwithpixels.com`

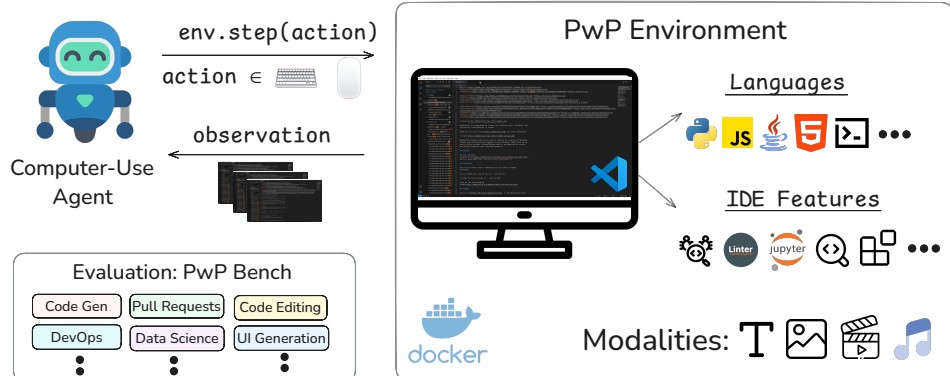

Figure 1: `Programming with Pixels` is an environment for computer-use agents, where they interact with a VSCode IDE through keyboard and mouse actions while observing the screen. The framework supports multiple programming languages, tests interactions with multiple IDE features, modalities (eg: text, images, data files). `PwP-Bench` evaluates agents across 15 diverse software engineering tasks such as code generation, UI generation, Data Science.

raises a fundamental question: can general-purpose computer-use agents match specialized agents in complex domains like software engineering?

To investigate this question, we introduce `Programming with Pixels` (PwP), the first environment for systematically evaluating computer-use agents on software engineering tasks. The `PwP` environment provides a VSCode-based IDE where agents perceive the screen and use primitive actions such as typing and clicking to perform a variety of SWE tasks. This design enables two critical properties. First, the environment is *expressive*, allowing agents to complete any software engineering task achievable in an IDE without language- or domain-specific modifications. Second, agents can access all IDE tools—debuggers, linters, code suggestions—through the same visual interface available to human developers or specialized SWE agents. Hence, `PwP` provides a general-purpose, realistic software engineering environment for testing computer-use agents.

To evaluate computer-use agents, we construct `PwP-Bench`, a benchmark of 15 tasks spanning different tasks such as code generation, pull request resolution, UI development, and data science across multiple programming languages and modalities. The benchmark represents a unification of 13 existing SWE tasks ported for evaluating computer-use agents, and 2 additional tasks developed by us. Our evaluation of state-of-the-art computer-use agents reveals that when restricted to pure visual interaction, these agents achieve only 22.9% average accuracy, significantly underperforming specialized coding agents. However, when augmented with just two basic text APIs—file editing and bash operations—the same agents achieve 50.7% accuracy, often approaching specialized agent performance despite their task-agnostic design. Furthermore, our analysis reveals substantial opportunities for future work. First, even state-of-the-art computer-use agents suffer from visual grounding issues. Second, we show that current computer-use agents lack the ability to use many of the tools available in the IDE, including ones that could make their tasks trivial. This suggests that training computer-use agents to explore and leverage the functionality present in their computer environment is a fruitful future direction. Overall, our results highlight software engineering as a realistic and challenging benchmark for evaluating and improving computer-use agents.

In summary, our contributions are as follows. First, we introduce `Programming with Pixels` (PwP), the first software engineering-focused environment for evaluating computer-use agents. Second, we introduce `PwP-Bench`, a benchmark spanning 15 diverse SWE domains, allowing for systematic comparison of computer-use agents. Third, through extensive evaluation, we highlight the limitations of current computer-use agents, identifying the need for models that have better visual grounding and that better take advantage of their environment as key future directions. Finally, we open-source our environment and benchmark, allowing it to serve as an open platform for evaluating and improving agents on software engineering tasks.

## 2 RELATED WORK

**Multimodal and Computer-Use Agents.** Recent works have explored using multimodal LLM agents to operate user interfaces such as web browsers (Koh et al., 2024; Deng et al., 2023; Zheng et al., 2024) and operating systems (Xie et al., 2024; Bonatti et al., 2024). Agent designs in these settings fall into two categories: (a) agents with predefined action sets (e.g., `new_tab`, `go_back`, `click [element id]`) that receive auxiliary information such as HTML accessibility trees (Yang et al., 2023a) for visual grounding; (b) pure computer-use agents operating with primitive keyboard and mouse actions, relying solely on screenshots (Anthropic, 2024; OpenAI, 2025; Qin et al., 2025). `PwP` supports evaluating both agent designs. Further, existing benchmarks such as OSWorld (Xie et al., 2024), AndroidWorld (Rawles et al., 2025), and WindowsAgentArena (Bonatti et al., 2024) evaluate agents on simple tasks like document editing and calendar management, leaving unclear whether performance on these tasks translates to complex, specialized domains like software engineering. `PwP-Bench` fills this gap by providing the first benchmark specifically designed to test whether computer-use agents can handle software engineering tasks. While some prior works explores specialized domains such as game playing (Tan et al., 2024) and a concurrent work explores scientific software (Sun et al., 2025), `PwP` is the first environment and `PwP-Bench` the first benchmark systematically evaluating computer-use agents for software engineering, a domain that is particularly noteworthy due to the presence of strong specialized agent baselines.

**Software Engineering Agents.** Software engineering agents have primarily relied on specialized scaffolding tailored to specific tools, languages, or tasks (Jin et al., 2024; Yang et al., 2024b). For instance, Agentless (Xia et al., 2024) uses Python-specific parsers, SWE-agent employs task-specific modifications (Abramovich et al., 2024; Yang et al., 2024b), and others depend on hand-engineered components like IPython kernels (Wang et al., 2024a) or custom browser views (Yang et al., 2024b). Our work takes a fundamentally different approach by evaluating whether computer-use agents – which interact through the same visual interface as human developers– can match these specialized agents. This also tests whether visual interaction with standard developer tools is sufficient for software engineering or if specialized APIs remain necessary. As `PwP` supports evaluating both designs, it enables direct comparison between computer-use and specialized agents across the diverse tasks in `PwP-Bench`, establishing a unified platform for understanding the capabilities and limitations of different agent designs. We refer readers to Appendix C for a more detailed related work.

## 3 PROGRAMMING WITH PIXELS (PwP)

Testing computer-use agents (CUAs) on software engineering (SWE) requires an environment that captures the full complexity of modern software engineering, which involves multiple programming languages, tools, and modalities. Furthermore, a fair evaluation must provide access to the wealth of tools that human developers use and specialized AI SWE agents have access to, such as linters, visual debuggers, and even project management tools. To enable such evaluation, we create `Programming with Pixels`, an IDE environment that satisfies these two requirements. First, it is *expressive*, meaning that an agent can perform any task that is achievable through a sequence of primitive operations (e.g., typing or clicking) within an IDE, which includes a wide range of software engineering activities. Second, an agent has access to any functionality implemented within the IDE, since using IDE functionality amounts to performing a sequence of primitive actions.

**PwP environment.** We represent the `PwP` environment as a partially observable Markov decision process (POMDP). We define the PwP POMDP $\langle S, A, O, T, R \rangle$ as follows. **S** is the *set of states* describing the IDE and the operating system (OS) context, including open files, active editor panels, and cursor positions. **A** is the *action space*, encompassing all possible keyboard and mouse events. The atomic actions in PwP are provided by the `xdotool` library (Sissel), which allows specifying all possible keyboard and mouse events in a simple syntax. The specific action space varies based on the agent setting, described in (§5). **O** is the *observation space*. The observation space varies based on the agent setting, described in (§5). **T** is the *transition function*. Actions like inserting a character typically lead to deterministic changes in the IDE state, whereas background processes can introduce stochasticity in timing and responses. **R** is the *reward function* that measures performance on a given task. For instance, after the agent finishes editing code to fix a bug, the environment can run a test suite on the updated files to compute a reward. Trajectories in `PwP` thus resemble real-

world development work: an agent can fix a bug in a repository, use a suggestion tool to help with writing code, or create documentation. The IDE and OS environment track changes, run tests and return reward signals. In addition, we discuss five key features of `PwP`.

**1. Expressive observation and action space.** `PwP` provides computer-use agents with an unrestricted environment where they can attempt any software engineering task achievable through an IDE's visual interface, as humans do. Unlike environments with predefined action sets, agents can navigate IDE menus visually, move cursors, and press keys to perform more complex actions.

**2. Full Spectrum of Developer Tools.** When evaluating computer-use agents on SWE tasks, it is imperative that they have a similar level of access to tools as specialized SWE agents, such as those with custom APIs for debuggers, linters, refactoring utilities, and more (Xia et al., 2024; Yang et al., 2024b). `PwP` provides all these tools through IDE's visual interface, creating a comprehensive test of whether CUAs can leverage the same rich functionality that specialized agents access through APIs.

**3. Multimodality and language agnosticism.** CUAs promise generality across tasks and domains. Software engineering spans many languages such as Python, Java, JavaScript, Lean, and more, with tasks involving multiple modalities, such as text, images, data files, and PDFs, providing a rigorous test of this generality. In `PwP`, the same CUA must handle code generation, UI development, data science, and theorem proving without task-specific modifications. For agents requiring visual grounding support, we modified VSCode's source code to provide rich DOM trees and Set-of-Marks annotations, ensuring fair evaluation across different CUA architectures.

**4. Ease of verification.** `PwP` provides direct access to the IDE's internal state, file system, and OS processes for verification. When an agent modifies code, we can run test suites, check compilation, and verify correctness. This separation between agent interaction (visual) and evaluation (programmatic) makes it easier to verify task completion and provide other sources of feedback.

**5. Future adaptability.** Computer-use agents are improving rapidly, and so are software engineering agents. `PwP` is designed for future adaptability. First, adding new benchmarks is as simple as modifying configuration files. Second, `PwP`'s checkpointing is useful for search and RL training methods. Third, `PwP`'s gymnasium interface (Towers et al., 2024) provides a standard interface for evaluation and development. Finally, as agents improve and become capable of using more complex tools, the environment (IDE) would automatically incorporate these without architectural changes. This makes `PwP` an extensible platform for evaluating and developing computer-use agents.

**Infrastructure and Implementation** `PwP` is deployed in a secure sandboxed docker environment, running open-source VSCode and a minimal operating system. Each container is isolated, preventing interference between experiments, ensuring parallel evaluation and facilitating reproducibility. We implement checkpointing for the environment state, which is especially useful for backtracking in search algorithms or training RL agents. The environment interfaces to VSCode using four channels for real-time screen capture, DoM information, and customizable configuration such as display, CPU/memory limits, etc. However, the complex interaction is abstracted away from the user, as they can simply interact with the environment through gymnasium python API (See Figure 9) and install the environment using a simple pip command. We refer to subsection A.3 for more details.

## 4 PwP-Bench

We introduce `PwP-Bench`, a benchmark containing 15 diverse software engineering tasks that span 14 programming languages and multiple modalities. Each task provides agents access to the IDE via the `PwP` environment. The goal of `PwP-Bench` is to test whether computer-use agents (CUAs) can handle the depth and breadth of software engineering activities.

**Tasks.** `PwP-Bench` contains 5400 instances sourced from 13 existing code-generation datasets and 2 newly created by us. These tasks are designed to be representative of software engineering activities that take place within an IDE. Since the IDE is simply a computer program, in principle, these activities should be achievable by a general-purpose computer-use agent. We selected the tasks in `PwP-Bench` according to three key principles: (1) tasks must require substantial interaction with software engineering tooling, (2) each task should require multiple steps, and (3) the benchmark must cover multiple languages and modalities. Accordingly, tasks are grouped into four categories:

- **Code Generation and Editing ($n = 6$):** These tasks evaluate the ability to generate and edit code. This category includes datasets such as HumanEval for code completion, SWE-Bench (Jimenez et al., 2023) and SWE-Bench-Multilingual (Yang et al., 2025) for resolving pull requests, DS-Bench for data science tasks (Jing et al., 2024), and Res-Q (LaBash et al., 2024) or CanITE-dit (Cassano et al., 2024) for code editing. Each dataset benefits from different IDE functionality. For example, SWE-Bench can take advantage of debuggers and linters, while DSBench may leverage an IPython kernel and extensions for analyzing large data files. Code editing tasks can leverage refactoring utilities and repository searches, covering varied input-output formats and end goals.

- **Multimodal Code Synthesis ($n = 4$):** These tasks involve creating code based on input images or other visual data. Examples include Design2Code (Si et al., 2024b) for UI development, Chart2Mimic (Shi et al., 2024) for generating Python code from chart images, SWE-Bench-MM (Yang et al., 2024b) for multimodal code editing, and DSBench tasks that rely on images, data files, or PDF documents for data analysis.

- **Domain-Specific Programming ($n = 3$):** These tasks focus on specialized fields such as ethical hacking (CTF) (Yang et al., 2023b) and interactive theorem proving (miniCTX) (Hu et al., 2024), which demand significant use and interaction with the IDE's functionality. For example, theorem proving requires continuously inspecting goal states via the IDE, while CTF tasks involve analyzing images, running executables, or installing VSCode extensions (e.g., hexcode readers).

- **IDE-Specific and General SWE Tasks ($n = 2$):** Since code generation is only one aspect of software engineering, we introduce two novel task sets that evaluate broader SWE skills. The first, **IDE Configuration**, evaluates an agent's ability to modify IDE settings such as themes, extensions, and preferences. These skills involve substantial interaction with the IDE, and are often a precondition for using IDE functionality such as new extensions. The second, which we term **General-SWE**, targets five different non-code activities: performance profiling, code refactoring, debugging bugs in standard libraries, UI mockup design, and code restoration. These tasks target practical software engineering tasks typically absent in conventional benchmarks. Full details are in Appendix B.2.

The distribution of tasks across categories and modalities is shown in Figure 10 in the Appendix. Computer-use agents that perform well across these tasks would demonstrate strong potential for automating diverse SWE activities across multiple languages, and working with varied input/output modalities such as text, images, data files, and other data types. Furthermore, taking advantage of the functionality provided by the agent's environment is essential.

**Benchmarking Design and Task Setup.** All tasks are evaluated within the `PwP` environment. Unlike traditional benchmarks, `PwP-Bench` presents agents with a realistic IDE environment: each agent receives an initial IDE state $S_i$ and an instruction $I$, with the goal to achieve a final state $S_f$ evaluated via execution-based criteria (e.g., unit tests). Among other capabilities, this setup tests whether CUAs can find relevant information from files, directories, and other resources, which is important for complex software development. Furthermore, a task is defined by a simple setup script that defines the initial IDE state, the instructions, and the evaluation logic. This makes it easy to add new tasks, allowing `PwP-Bench` to evolve as new benchmarks or better agents are developed.

**`PwP-Bench-Lite`.** Because `PwP-Bench` contains more than 5400 instances in total, running a full evaluation can be computationally expensive. To address this, we also provide `PwP-Bench-Lite`: a smaller subset of 300 instances. Specifically, we randomly sample 20 tasks from each of the 15 benchmarks. This subset preserves the overall difficulty and distribution while ensuring equal representation for each task, thereby making rapid experimentation more accessible.

## 5 EVALUATING AGENTS IN PROGRAMMING WITH PIXELS

We evaluate three distinct agent designs in the `PwP` environment to understand the capabilities and limitations of computer-use agents for software engineering tasks.

**Computer-use agents.** Computer-use agents interact with the IDE through primitive actions, i.e., keyboard and mouse inputs, while observing the interface visually through screenshots. Each agent operates in a turn-based manner, receiving a screenshot each turn and returning an action to progress toward the goal. Since most vision-language models without GUI-specific training struggle with raw

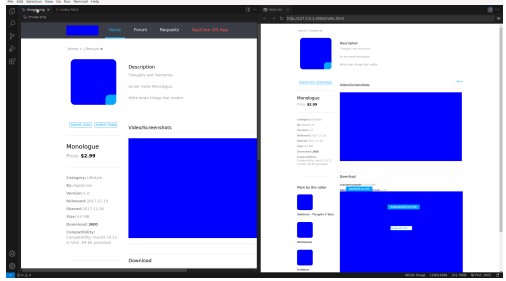

Figure 2: **Example of Successful Use of Live Preview Tool in the UI Replication Task** The agent successfully uses the live preview tool in the VSCode browser to compare the UI design it made versus the reference design.

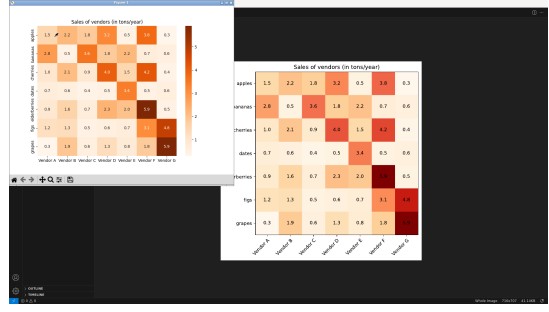

Figure 3: **Example of Successful Use of Tool in the Chart Generation Task** The agent can compare the generated chart with the reference chart side by side and refine its code accordingly.

pixel coordinates, we incorporate *Set-of-Marks (SoM)* (Yang et al., 2023a). With Set-of-Marks, an agent receives both the raw image and a parsed representation of available interface elements (e.g., buttons, text fields), allowing them to interact via element IDs rather than pixel coordinates. This design follows previous works (Xie et al., 2024; Koh et al., 2024).

**Computer-use agents with File/Bash APIs.** Computer-use agents are augmented with direct access to file-editing and bash commands through text APIs. The file-editing APIs include operations such as 'read file' and 'string replace', while bash operations allow command execution in the terminal. Agents receive screenshots only when requested via a screenshot action, rather than automatically each turn. This design strictly follows Anthropic's computer-use implementation (Anthropic, 2024).

**Specialized software engineering agents.** For comparing how well current computer-use agents perform relative to specialized agents, we evaluate mini-swe-agent (SWE-agent, 2024), an agent scaffold specifically designed for software engineering. Unlike computer-use agents that interact visually with the IDE, mini-swe-agent operates entirely through text APIs. For multimodal tasks, it receives required images directly as input in its prompt. We chose mini-swe-agent due to its near state-of-the-art performance on the widely-used benchmark SWE-Bench, as well as its flexibility for adapting to different programming tasks. See Appendix D.1 for implementation details and Appendix I for the prompts used for each agent.

**Experimental setup.** We test multiple models as the parametrization for the two computer-use agent designs. Specifically, we test four vision-language models: Gemini-Flash-1.5, Gemini-Pro-1.5, GPT-4o, GPT-4o-mini, and we test six models with UI-specific training: closed-source Claude-3.5 Sonnet, Claude-3.7 Sonnet, Claude-4.0 Sonnet, and open-weights Qwen-2.5-VL, Qwen-GUI-Owl-32B, and Qwen3-VL-30B-A3B. For mini-swe-agent (with multimodal support), we test Claude-4.0 Sonnet. We also evaluate a text-only version of mini-swe-agent on multimodal tasks by withholding image inputs to assess the importance of visual modality (see Appendix F). We keep the experimental setup consistent across all tasks and models: for each task instance, the maximum number of iterations is capped at 20 steps; if the agent either exhausts these steps or issues a stop command, the environment's final state is evaluated using task-specific metrics (see Appendix B.2 for full details). For SWE-Bench related tasks, we further evaluate with a maximum of 250 steps in Appendix F. Due to computational and budget constraints, we evaluate on `PwP-Bench`-Lite, which has 300 task instances.

## 5.1 RESULTS AND ANALYSIS

Table 1 summarizes performance across different agent architectures and base models over the four categories of `PwP-Bench` (task-wise results are in Table 8). As seen in the top half of the table, computer-use agents using only primitive keyboard and mouse actions achieve poor performance, with a maximum overall average of 22.9%. This is significantly lower than the software-engineering specific agent mini-swe-agent, which achieves 48.8% accuracy. We attribute this poor performance primarily to limited visual grounding and an inability to interact effectively with the IDE, particularly

Table 1: Performance Evaluation of Different Agents on `PwP-Bench`-Lite by Task Categories. Best numbers are in bold, and best numbers for computer-use agents are underlined.

| Model | Code Generation & Editing (n=6) | Multimodal Code Gen. (n=4) | Domain-Specific Code Gen. (n=3) | General SWE Tasks (n=2) | Overall Avg |
|---|---|---|---|---|---|
| *Computer-Use Agents* | | | | | |
| Gemini-Flash | 0.0% | 4.3% | 0.0% | 0.0% | 1.1% |
| GPT-4o-mini | 0.8% | 3.7% | 0.0% | 2.5% | 1.7% |
| Qwen2.5-VL-72B | 0.0% | 4.3% | 0.0% | 5.0% | 1.8% |
| GUI-Owl-32B | 0.0% | 0.0% | 0.0% | 22.5% | 3.0% |
| Qwen3-VL-30B-A3B | 0.8% | 9.0% | 0.0% | 37.5% | 7.7% |
| Gemini-Pro | 2.5% | 5.7% | 0.0% | 7.5% | 3.5% |
| GPT-4o | 0.8% | 12.4% | 1.7% | 10.0% | 5.3% |
| Claude-Sonnet-3.5 | 10.7% | 8.3% | 5.0% | 22.5% | 10.5% |
| Claude-Sonnet-3.7 | 11.8% | 28.5% | 8.3% | 27.5% | 17.7% |
| Claude-Sonnet-4.0 | $14.3_{\pm1.2}$% | $37.3_{\pm0.6}$% | $6.7_{\pm0.0}$% | $40.0_{\pm3.5}$% | $22.3_{\pm0.5}$% |
| *Computer-Use Agents with File/Bash APIs* | | | | | |
| Gemini-Flash | 9.5% | 11.7% | 8.3% | 2.5% | 8.9% |
| GPT-4o-mini | 23.6% | 17.6% | 15.0% | 5.0% | 17.8% |
| Qwen2.5-VL-72B | 13.7% | 11.8% | 6.7% | 7.5% | 11.0% |
| Gemini-Pro | 30.0% | 16.7% | 3.3% | 12.5% | 18.8% |
| GPT-4o | 36.2% | 41.9% | 28.3% | 10.0% | 32.6% |
| Claude-Sonnet-3.5 | 47.9% | 55.1% | 43.3% | 22.5% | 45.5% |
| Claude-Sonnet-3.7 | 51.9% | 58.7% | **46.7%** | 27.5% | 49.4% |
| Claude-Sonnet-4.0 | $53.5_{\pm0.2}$**%** | $57.8_{\pm1.4}$% | $43.9_{\pm2.0}$% | $38.3_{\pm1.2}$**%** | $50.7_{\pm0.2}$**%** |
| *Software Engineering Agents* | | | | | |
| mini-swe-agent | 49.4% | **60.3%** | 40.0% | **37.5%** | 48.8% |
| OpenHands | 50.4% | 50.8% | 43.3% | 25.0% | 45.7% |

for file editing and tool usage; see Section 5.1 for further analysis. Among all evaluated models, the Claude computer-use agent performs best, likely because it is specifically trained for UI interactions. We found that it can leverage basic IDE tools such as HTML live preview, chart visualization, and file navigation, boosting performance on tasks that require visual understanding and IDE navigation.

Nonetheless, when the same computer-use agents are granted access to just two text APIs (file editing and bash operations) we observe consistent improvements across all categories, with the maximum average accuracy reaching 50.7

However, models still struggle to fully leverage the tooling available in the IDE. This is evidenced by poor performance on the 'General SWE' category, where tasks often require fewer than ten steps when using appropriate IDE tools. We analyze the poor performance on General SWE tasks further in the following sections, confirming that these tasks would become simpler if models could use IDE tooling more effectively. Overall, our results show that computer-use agents to have some facility for software engineering, but currently require better visual grounding, tool usage, and planning. In the following paragraphs, we analyze these strengths and deficiencies in more detail.

**Claude Computer-Use Agent Demonstrates Basic IDE Tool Proficiency.** Qualitatively, we found that Claude Computer-Use agent can use basic IDE functionalities, including file explorer navigation, file editing, search, browser-based live preview, and image generation and visualization capabilities. Figure 2 demonstrates the agent's effective use of browser tools in UI replication tasks. Similarly, Figure 11 illustrates the agent's ability to utilize multiple tools while editing specific lines in a repository, relying solely on screenshot observations and primitive keyboard/mouse actions.

Furthermore, we hypothesize that agents have additional latent abilities to use tools that can be activated through prompting or fine-tuning. To investigate this, we examined the project refactoring task (such as symbol renaming) in our 'General-SWE' benchmark, where Claude initially achieves 25% accuracy when attempting the task. However, when explicitly instructed to use precise tools (such as rename or move to file), its accuracy improves to 75% (see Appendix E).

**Computer-Use Agents Demonstrate Poor Visual Grounding Capabilities.** While, Claude Computer-Use agent is able to use basic IDE tools, we found that in general all current CUAs have

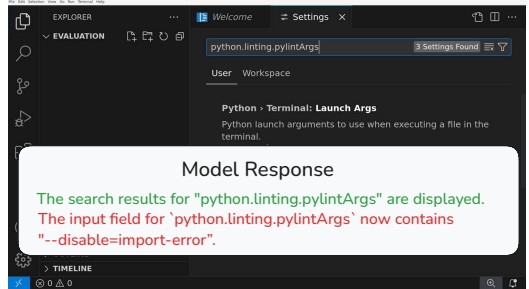 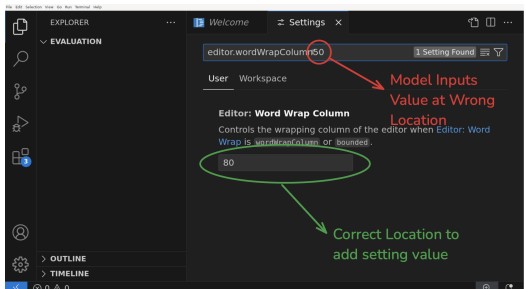

Figure 4: **Agent Hallucinating Screen Contents** The agent correctly mentions, search results are displayed (green), it hallucinates an input field containing "disable import error" (red).

Figure 5: **Agent Misidentifying UI Elements** The agent fails to identify the correct input field, typing '50' into the settings search bar instead of the word wrap column setting field (red arrow).

significant limitations in visual grounding, i.e., the ability to understand the visual input and take actions on the visual IDE interface. We identify three primary failure modes. First, the agents can often fail to use the correct UI elements. For example, in Figure 5 the agent types in the search bar rather than the settings field, while in Figure 15 the model clicks the wrong location. Surprisingly, Set-of-Marks did not resolve these issues; agents would instead select incorrect elements.

Second, the agents often struggle to comprehend the current UI state, such as linter errors indicated by wavy underlines (Figure 13) or hallucinate screen contents (Figure 4). Finally, even when the agent can identify a simple error, such as incorrect indentation, it is often not able to fix the error due to struggling with clicking and typing in the proper locations. Furthermore, in Appendix F, models frequently completely ignore the visual state information and instead rely on completely memorized action sequences. Quantitatively, we found that 20% and 95% of trajectories have at least one visual grounding error in GPT-4o and Claude Sonnet-4.0, respectively (see Appendix F).

While grounding has been highlighted as a weak point of computer-use agents in web and OS domains (Koh et al., 2024; Xie et al., 2024), the limitations were primarily observed in models without UI-specific training. However, our work shows that even models explicitly trained for UI interaction, such as Claude Computer Use (Anthropic, 2024), exhibit these issues in `PwP`. We hypothesize that the deficiencies come from the IDE being particularly information-dense, as well as potentially not being covered by computer-use training datasets.

**Agents Struggle to Use Advanced IDE Functionality.** Although the best computer-use agent we tested could use basic IDE functionality, all agents lack the ability to leverage more sophisticated IDE tools. Specifically, we can see this through the low performance on the 'General-SWE' dataset, which focuses on software engineering activities (e.g., profiling, refactoring, debugging) that can be often completed without direct code edits. Although these tasks sometimes require only 4-5 steps when using appropriate IDE tools, agents achieve minimal performance, highlighting substantial room for improvement. Furthermore, we observed no successful uses of profilers, debuggers (even when explicitly instructed to) when performing the other tasks in our benchmark (see Appendix E). We further quantify the IDE features used across all trajectories of Claude-Sonnet-4 CUA in Appendix F.

**Distribution of Functionality Used by Computer-Use Agents with File/Bash Operations.** As we observed in Table 1, computer-use agents perform much better when they have access to file and bash API calls, which are based on text inputs and text outputs. A natural question is to what extent these are using the visual interface versus relying on text-only APIs. We study this in Figure 6, which shows the distribution of file/bash API calls versus computer-use interactions on four representative datasets. The figure shows a few interesting patterns. First, for HumanEval, agents rely entirely on file APIs. This is because HumanEval tasks involve simple function completions that are achievable without IDE interaction. The lower performance of pure CUAs on this task (25% compared to 100%) demonstrates their inability to perform basic file editing visually. Second, for SWE-Bench-MultiModal, surprisingly there are minimal computer-use interactions, primarily using screenshots to understand the open repository or occasionally attempting to open the built-in browser.

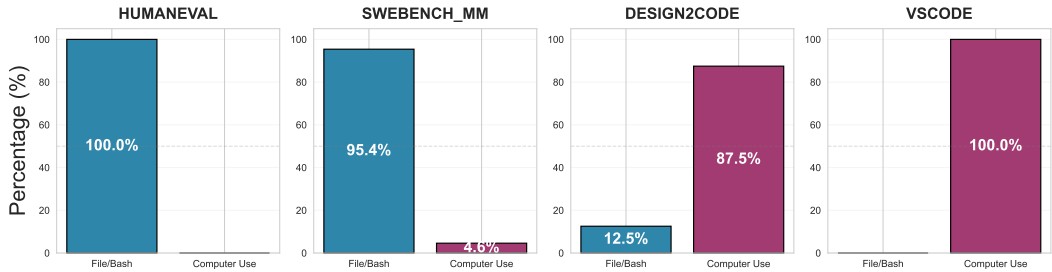

Figure 6: Distribution of file/bash calls vs computer-use interaction for computer-use agents.

In contrast, the distribution shifts dramatically for Design2Code, where agents frequently open live preview tools to compare generated designs with reference images, and continuous refining the output (see Figure 2). In a similar vein, for VSCode tasks, the agents rely entirely on visual IDE functionality to update settings, install extensions, and edit themes. These patterns demonstrate that computer-use agents with file/bash APIs have some ability to choose between visual and API based interactions based on the task requirements. On datasets such as HumanEval, their performance improvements stem from bypassing their inability to visually perform edits, instead using text APIs.

**Computer-Use Agents Are Rapidly Improving.** Figure 7 compares the performance of Claude-Sonnet 3.5, 3.7, and 4.0 released over a period of 7 months. The line shows steady improvement in pure CUAs, with performance nearly doubling from 10.5% to 22.9%. Furthermore, from Table 1 we see that the gap between pure CUAs and CUAs with file/bash operations has steadily decreased from 35.0% to 27.8%. These results highlight that while a substantial gap remains, rapid progress is being made and continued improvements may eventually close this gap.

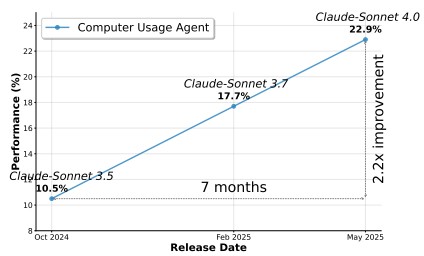

Figure 7: Performance of Claude-Sonnet Computer-Use Agents over time

**Leveraging the IDE functionality better would improve performance.** While a single computer-use agent design can perform non-trivially across a wide variety of tasks, our analysis indicates that these models do not fully exploit domain-specific tools. To quantify the potential performance gains if agents could effectively use the IDE, we perform an "assisted" experiment. In this experiment, we manually engineered a set of IDE-based tool calls representing commonly used IDE functionalities (e.g., live HTML previews, repository structure, symbol outlines). Importantly, each API call is achievable using basic operations in the IDE, meaning that in principle, an agent could learn to perform it. See Appendix E for full details.

Table 8 summarizes the performance improvements of assisted agents, highlighting an average gain of up to 13.3%. These results demonstrate that current CUAs have poor interaction capabilities with complex interfaces, yet there is significant scope for improvement. The results also suggest that

Figure 8: Assisted versus Computer-Use Agents

|  | SWE-Bench | Design2Code | Chartmimic | BIRD (T2 SQL) |
|---|---|---|---|---|
| Computer Use Agents | 0% | 23.5% | 2.7% | 0% |
| CUA + File/Bash | 15% | 48.1% | 25.3% | 7% |
| Assisted | **19%** | **79.5%** | **61.6%** | **17%** |

in the near term, performance gains can be achieved by introducing specialized hand-engineered tools into computer-use agents and incorporating existing agent designs in our `PwP` environment.

## 6 CONCLUSION

We introduce `Programming with Pixels`, an environment designed to evaluate computer-use agents on software engineering tasks. We also introduce `PwP-Bench`, a diverse benchmark of 15 tasks spanning the breadth of software engineering across multiple languages and modalities. Our extensive evaluations of nine models reveal that pure computer-use agents relying solely on visual interaction perform poorly, while augmenting these agents with simple file and bash text

APIs dramatically improves performance. Our analysis pinpoints poor visual grounding and an inability to leverage the rich set of functionality in the `PwP` environment as primary weaknesses. Despite these limitations, our findings show that CUAs are improving rapidly, signaling significant potential. `PwP` establishes software engineering as a natural domain for benchmarking whether generalist computer-use agents can reach specialist-level performance on sophisticated tasks.

## 7  ACKNOWLEDGEMENTS

We thank Daniel Fried, Graham Neubig, Zora Wang, Saujas Vaduguru, Atharva Naik, Riyaz Ahuja, and Weihua Du for helpful feedback. We thank Convergent Research and the OpenAI Researcher Access program. This work was supported in part by the National Science Foundation under Grant Nos. DMS-2434614 and DMS-2502281. Pranjal is supported by SoftBank Group–Arm Fellowship.

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

Table 2: Comparison of Hand-engineered Tools across Methods versus `PwP`. `PwP` natively supports all tools.

| Method | Hand-engineered Tools | Supported in `PwP` |
|---|---|---|
| Agentless (Xia et al., 2024) | File Edit, Repository Structure, File Structure | ✓ |
| CodeAct (Wang et al., 2024a) | File Edit, IPython, Bash | ✓ |
| SWE-agent (Yang et al., 2024a) | Search File, Search Text, File Edit | ✓ |
| EnIGMA (Abramovich et al., 2024) | SWE-agent Tools + Debugger, Terminal, Connection Tool | ✓ |
| swebench-mm (Yang et al., 2024b) | SWE-agent Tools + View Webpage, Screenshot, Open Image | ✓ |

Table 3: Comparison of different environments across multiple dimensions

| Environment | Computer-Use Environment? | Execution-Based Reward | Specialized Domain | SWE Specific |
|---|---|---|---|---|
| GAIA (Mialon et al., 2023) | ✗ | ✗ | ✗ | ✗ |
| WEBSHOP (Yao et al., 2023) | ✗ | ✗ | ✗ | ✗ |
| WEBARENA (Zhou et al., 2024) | ✗ | ✓ | ✗ | ✗ |
| VWEBARENA (Koh et al., 2024) | ✓ | ✓ | ✗ | ✗ |
| BrowserGym (Chezelles et al., 2024) | ✓ | ✓ | ✗ | ✗ |
| OSWORLD (Xie et al., 2024) | ✓ | ✓ | ✗ | ✗ |
| AndroidWorld (Rawles et al., 2025) | ✓ | ✓ | ✗ | ✗ |
| WindowsAgentArena (Bonatti et al., 2024) | ✓ | ✓ | ✗ | ✗ |
| ScienceBoard* (Sun et al., 2025) | ✓ | ✓ | ✓ | ✗ |
| Cradle* (Tan et al., 2024) | ✓ | ✓ | ✓ | ✗ |
| `PwP` (Ours) | ✓ | ✓ | ✓ | ✓ |

# A  PROGRAMMING WITH PIXELS (`PwP`) ENVIRONMENT

## A.1  TOOLS

Previous methods have proposed use of various hand-engineered tools. For a fair comparison, all tools should be accessible in the `PwP` environment. Aas shown in Table 2, `PwP` natively supports all these tools.

## A.2  COMPARISON WITH OTHER ENVIRONMENTS

In Table 3, we compare `PwP` with existing environments across multiple dimensions. We evaluate environments along the following dimensions:

- **Computer-use environment**: Whether the environment is designed for computer-use agents, and thereof whether it supports multimodal interaction.

- **Execution-based evaluation**: Use of runtime execution to verify the correctness of agent actions

- **Specialized**: Whether the environment is designed for general and basic tasks, such as web navigation, or is it designed for a more sophisticated, specialized and potentially economically important tasks. Only Cradle (Tan et al., 2024) and ScienceBoard (Sun et al., 2025) are specialized for Game Playing and using Scientific softwares respectively.

- **SWE-specific**: Whether the environment is purposefully designed for software engineering tasks

Further, ours support other engineering features that others do not. For instance, `PwP` also support streaming video and audio, something other environments do not support out of the box. Further, unlike environments such as OS-World, which require manual creation of environment image, `PwP` is natively docker based, and is based on simple scripts, that can be easily used to modify startup scripts and other configurations for future adaptations. Finally, we also specifically suppport state checkpointing which supports storing file system and complete process state, and is especially useful for search-based methods.

```
1  bench = PwPBench(dataset='swebench')
2  # Replace with any dataset from PwP-Bench
3  dataset = bench.get_dataset()
4
5  # Set up environment and get initial observation
6  env = bench.get_env(dataset[0])
7  observation: PIL.Image = env.get_observation()['screenshot']
8
9  # Generate and execute action
10 action = agent.get_action(observation)
11 print(action)
12 # Output: xdotool mousemove 1000 1200
13 # click 1 && xdotool type 'hello world'
14 observation, info = env.step(action)
15
16 env.render()
17
18 # Environment control
19 env.pause()
20 env.resume()
21
22 # Get reward and reset
23 is_success = env.get_reward()
24 env.reset()
25
```

Figure 9: Example demonstrating interaction with PwP environment, including keyboard/mouse actions, checkpointing, and state management. The code shows basic initialization, action execution, environment control, and reward handling.

### A.3 INFRASTRUCTURE AND IMPLEMENTATION

`PwP` is deployed in a secure sandboxed environment. In particular, we run a modified version of Visual Studio Code (VSCode) and a minimal operating system inside a Docker container, ensuring a secure and isolated environment. We chose VSCode for its extensive language support, rich ecosystem of extensions, widespread adoption in the developer community, and open-source nature that enables customization and modification of its core functionality. Each container instance maintains its own file system and processes, preventing interference between experiments, facilitates reproducibility, and ensuring parallelization of evaluation. We further provide the ability to checkpoint the environment state, which is especially useful for backtracking in search algorithms or while training RL agents.

The environment interfaces with VSCode through multiple channels: 1.) A controller that manages Docker container lifecycle and configuration, 2.) A port-forwarding system for real-time screen and video capture, 3.) A modified VSCode codebase that exposes DOM state information, and 4) The VSCode Extension API for accessing fine-grained IDE state. This multi-channel approach enables both high-level environment control and detailed state observation.

Screen capture is handled via `ImageMagick` for static screenshots and `ffmpeg` for streaming video output. These tools were selected for their low latency and ability to handle various screen resolutions and color depths. For actions, a lightweight controller executes `xdotool` commands within the container, which in turn simulates keyboard and mouse events on the IDE. Agents can thus insert code, open new files, or navigate menus using the same actions that a human developer would.

As shown in Figure 9, a Python API is provided for interaction, following a style similar to common reinforcement learning libraries such as gymnasium (Towers et al., 2024). The API abstracts away the complexity of container management, benchmark management, and handling observations and actions, allowing researchers to focus on agent development. Users can query the environment for the latest screenshot, issue an `xdotool` command, and receive updated states or rewards. Examples

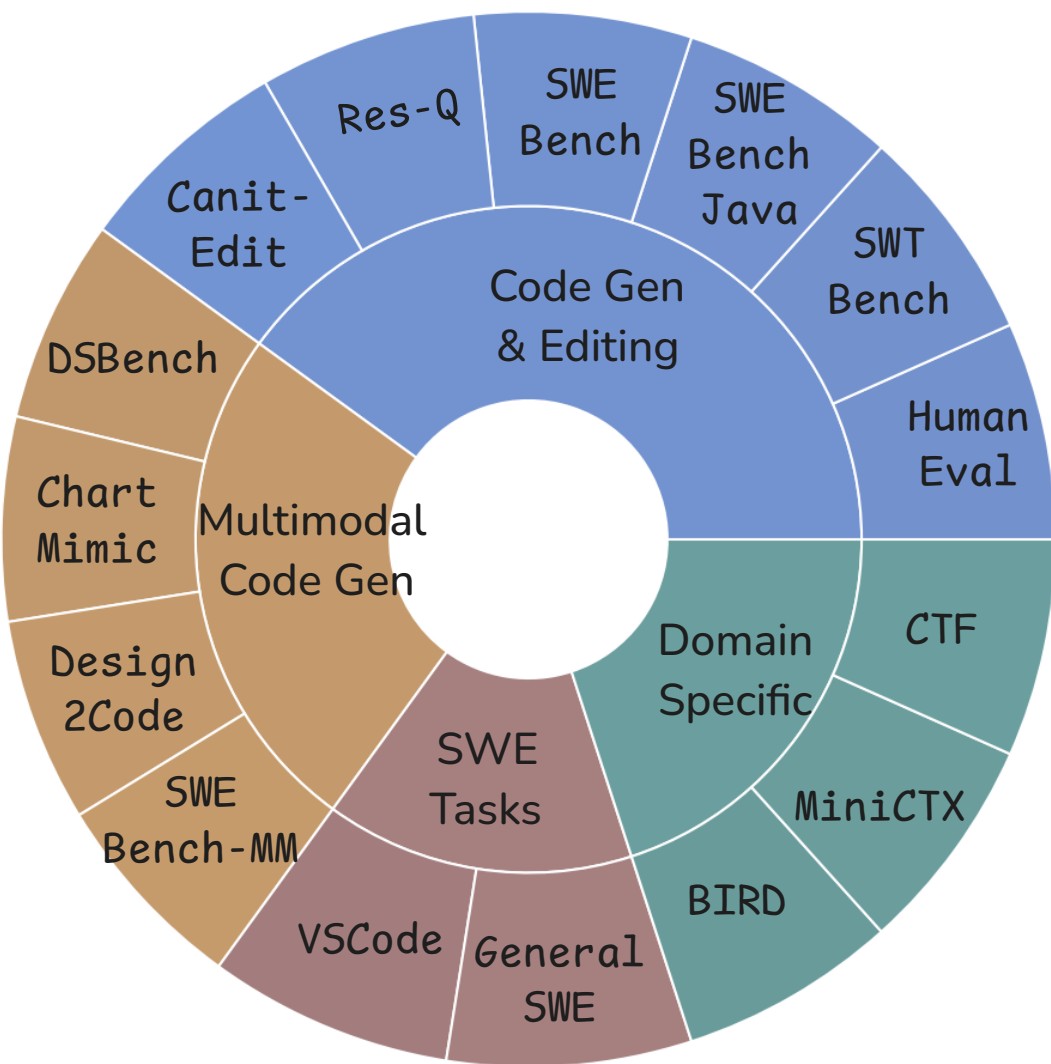

Figure 10: Distribution of tasks in PWP-Bench across four main categories: Code Generation and Editing, Multimodal Code Synthesis, Domain-Specific Programming, and General SWE Tasks. The inner ring shows the main categories while the outer ring shows specific datasets and tasks within each category. Note that the figure is not drawn based on relative size of tasks.

of xdotool commands include 'xdotool mousemove 1000 1200' and 'xdotool type 'hello world'' and are shown in Figure 9. The environment's container configuration is flexible, allowing for software installations, customizable CPU/memory limits, and display settings (e.g., resolution). This versatility is crucial for large-scale evaluation, especially when tasks vary in complexity and resource needs. Finally, the environment has been tested on three different operating systems: Ubuntu, MacOS, and Windows.

## B PWP-BENCH

### B.1 TASKS

Figure 10 shows the set of tasks across all categories. Further, Table 4 shows the number of instances for each task in the full benchmark, along with the languages used in each of the tasks. PwP-Bench-Lite contains 300 instances, which is a random sample of 20 instances from each task.

Table 4: Number of instances for each task in `PwP-Bench`

| Task | Number of Instances | Languages |
|---|---|---|
| HumanEval | 165 | Python |
| Design2Code | 485 | HTML/CSS/JS |
| ChartMimic | 600 | Python |
| InterCode | 100 | Python, Bash |
| RES-Q | 100 | Python |
| CanItEdit | 105 | Python |
| VSCode | 20 | - |
| Bird | 500 | SQL |
| DSBench | 112 | Python |
| SWE-bench | 2000 | Python |
| SWE-Bench-Multilingual | 91 | C++, Typescript, Javascript, Rust, Go, C, Ruby, PhP, Java |
| Swebench-MM | 510 | Javascript |
| SWT-Bench | 276 | Python |
| Minictx | 381 | Lean |
| General SWE | 20 | - |

## B.2 EVALUATION

All tasks are evaluated using programmatic verifiers. These verifiers are typically run on an separate environment, not accessible to the agent. This typically works based on fetching relevant files and information from the agent environment, and then running through task-specific evaluation scripts on a separate environment. However, to the user, this is abstracted away, and they simply have to call 'env.get_reward()' to get the exact score or correctness signal based on task.

**Metrics**  We use individual metrics mentioned in the original datasets. When reporting results on `PwP-Bench`, we report marco average of all these metrics. In particular, 11/15 used Accuracy as their metric. However, due to complexity of dataset, these often goes beyond simple accuracy metrics and in some cases, the dataset is evaluated on multiple orthogonal metrics, instead of one. We detail, these metrics for each of the datasets.

- **SWT-Bench** evaluates generated tests by the agent, and reports 6 different metrics: Applicability, Success Rate, F- X, F- P, P- P, and Coverage. We report the average of all 6 metrics.

- **ChartMimic** evaluates generated code on various metrics such as accuracy of text, colors used, legend etc. We average all metrics similar to the original dataset.

- **Design2Code** evaluates generated code on various metrics such as accuracy of text, position, clip score, etc. We average all metrics similar to the original dataset.

- **DSBench** has two categories, one containing MCQ questions, while the other containing generating code for Kaggle Competitions. We use 10/10 instances from each category in `PwP-Bench`-Lite. While MCQ questions are evaluated using Accuracy, the code generation part is evaluated using linear normalization between the baseline score (of the competition) and the score of the winner of competition.

**VSCode and General SWE Tasks**  In this section, we detail the VSCode and General SWE tasks in `PwP-Bench`, created by us. The VSCode tasks are mostly designed to evaluate the ability of agents to use basic VSCode features, such as renaming all instance of a symbol in file, installing extensions, changing themes, modifying specific settings. All these tasks are evaluated based on final IDE state, either by invoking the 'code' cli tool, configuration files stored in environment filesystem, or through direct access to VSCode state provided by `PwP` (see  subsection A.3). General-SWE tasks, involves 5 categories of tasks: 1.) QA based on code profiling (evaluated based on final answer by model which requires using appropriate profiling tools), 2.) code refactoring (assessed through automated tests on the final repository state), 3.) debugging bugs in standard libraries (evaluated based on the correctness of final code state), 4.) UI mockup design (assessed using CLIP scores), and 5.) code restoration, where the agent leverages VSCode's timeline feature to recover corrupted codebases, evaluated by the correctness of the restored state.

**Full List of Tasks**    We provide the full list of tasks for IDE Configuration (VSCode) and General-SWE below.

Table 5: List of VSCode Configuration Tasks

| ID | Task Description |
|----|-----------------|
| 1  | Install the pylance extension in VS Code. |
| 2  | Please help me change the background of VS Code to the photo background.jpg. |
| 3  | Change VS Code's color theme to Solarized Dark. |
| 4  | Please modify VS Code's settings to disable error reporting for Python missing imports. |
| 5  | Please help me open the autosave feature of VS Code and delay AutoSave operations for 500 milliseconds in the VS Code setting. |
| 6  | Please help me modify VS Code setting to hide all 'pycache' folders in the explorer view. |
| 7  | Can you delay VS Code autoSave for 1000 milliseconds? |
| 8  | Please help me configure VS Code settings so that "Format On Save" is enabled specifically for **Python** files, but disabled for all other file types. |
| 9  | Please modify the "Files: Exclude" setting to hide all files ending in '.log' and '.tmp' from the Explorer view, ensuring they don't clutter the workspace. |
| 10 | Please help me create a standard 'launch.json' configuration file for a Flask application within the current workspace, setting the port to 5000. |
| 11 | Please help me select the Python interpreter located at './venv/bin/python' for the current workspace, rather than using the system default. |
| 12 | Please help me search for and install a specific color theme extension called "Dracula Official", then immediately activate it after installation. |
| 13 | Please help me set a **conditional breakpoint** on line 45 of 'data_processor.py' that only pauses execution when the variable 'retry_count ¿ 3'. |
| 14 | Please help me configure VS Code so that on startup it reopens the last used workspace. |
| 15 | Please help me change all the places in this document that say "text" to "test" and "test" to "text". |
| 16 | Please help me remove the shortcut "Ctrl+F" for Tree view Find in the Explorer, and then assign "Ctrl+Alt+F" as the new shortcut for Tree view Find to avoid conflict with editor search. |
| 17 | Please modify VS Code's settings to disable error reporting for Python missing imports. |
| 18 | Please configure the suggestion list so that Code Snippets always appear at the very top of the suggestion list, above standard variable names or keywords. |
| 19 | Please help me install the Black Formatter extension and configure it as the default formatter for all Python files in this workspace. |
| 20 | Please install flamegraph extension in VS Code. |

**Comparison with Other Benchmarks**    In Table 7, we further compare `PwP-Bench` with other existing benchmarks.

## C    RELATED WORK

### C.1    COMPARISON TO SOFTWARE ENGINEERING AGENTS

**Task-specific SWE benchmarks**    Early neural code generation approaches were typically evaluated on fixed input-output pairs—for example, generating code from docstrings (Chen et al., 2021) or from general textual descriptions (Austin et al., 2021). Subsequent benchmarks extended these evaluations to interactive settings, such as resolving GitHub pull requests or writing unit tests for real-world code repositories (Jimenez et al., 2023; Zan et al., 2024; Mündler et al., 2025). More

Table 6: List of General SWE Tasks

| Category | Task Description |
|---|---|
| **Timeline** | Use the Timeline view to find the local history version of main.py where import requests was deleted ( 30 mins ago) and restore it.
Identify the Timeline entry immediately before the formatting action that changed spaces to tabs, and revert the file to that state.
Locate the 'Git: Staged Changes' entry in the Timeline and copy the validate_user function from that version into the current file.
Use the Timeline to identify when timeout changed from 5000 to 10000 in utils.js and revert to the version immediately prior. |
| **Profiling** | Analyze the generated flame graph to identify the innermost C-level function consuming the most CPU time during the NumPy random generation phase.
Use the flame graph to determine the execution time ratio between the compute_heavy function and the io_save function.
Locate the widest bar at the top of the stack trace (the 'tip') and identify which specific Python library call it corresponds to.
Identify the deepest stack level in the graph where the application spends at least 50% of its total execution time. |
| **Refactoring** | Rename the DataProcessor class to LegacyDataProcessor globally, ensuring that occurrences within comments and string literals are excluded from the update.
Swap all assignments and references of variables width and height.
Move the AuthHandler class from main.py to a new file named auth_utils.py, making sure all import references across the workspace are updated.
Rename the random class to pseudo_random across the whole workspace. |
| **Mockups** | Replicate the image exactly into the currently open Draw.io canvas using standard flowchart shapes.
Replicate the image exactly into the currently open Draw.io canvas using standard flowchart shapes.
Replicate the image exactly into the currently open Draw.io canvas using standard flowchart shapes.
Replicate the image exactly into the currently open Draw.io canvas using standard flowchart shapes. |
| **Debugging** | Trace the execution flow through the Middleware class to find the exact method call that is silently modifying the request.headers dictionary before it reaches the endpoint.
The script fails during the np.dot operation; use the debugger to inspect the internal array shapes inside the function call and identify the dimension mismatch.
We have some numpy code which isn't working correctly due to an issue inside a library function. Debug the code line-by-line inside the library function's source to identify the problematic area.
The code is failing specifically at the np.dot operation. Use the debugger to inspect the runtime values of the matrices involved to figure out the exact dimension mismatch issue. |

recently, efforts have broadened the scope of code generation to include multimodal tasks, where vision models must interpret images to generate correct code or edits (Si et al., 2024b; Shi et al., 2024; Jing et al., 2024; Yang et al., 2024b). However, each of these benchmarks is confined to specific languages, modalities, or task types. In contrast, our proposed `PwP-Bench` unifies these diverse evaluations into a single framework, encompassing multimodal and multilingual challenges that require interaction with a broad suite of IDE tools. Using this unified approach we reproduce the performance of established benchmarks and encourage the development of general-purpose agents capable of handling a variety of new software engineering tasks. We further compare our work with previous efforts in Tables 3 and 7.

**Software Engineering (SWE) Agents** Recent work has explored "code agents" that move beyond single-step neural code generation toward interactive methods, where intermediate feedback from tools informs subsequent actions. However, many of these approaches specialize in particular tools

Table 7: **Comparison of existing software engineering benchmarks**. `PwP-Bench` provides the largest dataset (5400 instances) and uniquely covers all aspects: multiple languages and modalities, real IDE interaction, interactive coding, and both code generation and general software engineering tasks.

| Benchmark | #Instances | Multiple Languages | Multiple Modalities | Real IDE Env | Interactive Coding | Non-Code SWE Tasks | Code-Generation SWE Tasks |
|---|---|---|---|---|---|---|---|
| SWE-Bench (Jimenez et al., 2023) | 2K | ✗ | ✗ | ✗ | ✓ | ✗ | ✓ |
| SWE-Bench-MM (Yang et al., 2024b) | $\leq 1K$ | ✗ | ✓ | ✗ | ✓ | ✗ | ✓ |
| LiveCodeBench (Jain et al., 2024) | $\leq 1K$ | ✗ | ✗ | ✗ | ✓ | ✗ | ✓ |
| Aider Polyglot (Aider, 2024) | $\leq 1K$ | ✓ | ✗ | ✗ | ✓ | ✗ | ✓ |
| TheAgentCompany (Xu et al., 2024) | $\leq 1K$ | ✗ | ✓ | ✗ | ✓ | ✓ | ✗ |
| VisualWebArena (Koh et al., 2024) | $\leq 1K$ | ✗ | ✓ | ✗ | ✗ | ✗ | ✗ |
| OSWORLD (Xie et al., 2024) | $\leq 1K$ | ✗ | ✓ | ✓ | ✗ | ✓ | ✗ |
| WindowsAgentArena (Bonatti et al., 2024) | $\leq 1K$ | ✗ | ✓ | ✓ | ✗ | ✓ | ✗ |
| `PwP-Bench` (Ours) | 5.4K | ✓ | ✓ | ✓ | ✓ | ✓ | ✓ |

or programming languages (Jin et al., 2024; Yang et al., 2024b), limiting their broader applicability. For example, Agentless (Xia et al., 2024) relies on a tool that parses files into Python-specific class and function structures. This fails to perform well in other languages or settings (Yang et al., 2024b) without manual modifications. Similarly, the SWE-agent requires modifications to adapt to different tasks (Abramovich et al., 2024; Yang et al., 2024b). In contrast, agents designed for `PwP` are inherently task and language-agnostic due to the expressive action and observation spaces mandated by our environment. Moreover, the diverse tasks in `PwP-Bench` require agents to generalize across a wide range of SWE challenges rather than excel in one narrowly defined area such as resolving pull requests.

Many existing agents also depend on hand-engineered tools that require human effort to implement and are susceptible to bugs. For instance, Agentless (Xia et al., 2024) leverages tools for parsing files into Python-specific structures; CodeAct relies on an IPython kernel (Wang et al., 2024a); SWE-Agent uses dedicated search and file editing tools (Yang et al., 2024a); AutoCodeRover requires a linter (Zhang et al., 2024); SWE-Agent EnIGMA develops specialized tools for CTF-style competitions (Abramovich et al., 2024); and SWE-Bench-MM (Yang et al., 2024b) implements a browser view. In `PwP`, these tools are inherently available within the IDE (as detailed in Table 9), and the agent's task is to effectively use them rather than being explicitly guided on which tool to use for each specific task.

Finally, current approaches often blur the line between the agent and the environment, as each agent is designed with its own specified action and observation spaces within a self-created environment. `Programming with Pixels` addresses this issue by unifying existing environments into a single, general-purpose platform on which agents operate. This clear separation of environment design from agent design standardizes evaluation and also allows any existing agent to be modeled within our framework, making it an important testbed for both current and future SWE agents.

## C.2 COMPARISON TO GENERAL VISUAL AND COMPUTER-USE AGENTS

**Visual Agents and Computer-Use Agents** A family of recent multimodal agent benchmarks require agents to operate user interfaces using a predefined, limited set of actions (e.g., `new_tab`, `go_back`, `click [element id]`) (Koh et al., 2024; Deng et al., 2023; Zheng et al., 2024) . These *visual agents* typically rely on additional prompting—such as set-of-marks techniques that supply an HTML accessibility tree containing textual and positional information—to overcome their inherent poor visual grounding capabilities (Yang et al., 2023a). Despite such aids, these agents often fail when faced with the complex and dense IDE interfaces found in our environment.

A separate family of *computer-use agents* (Anthropic, 2024; OpenAI, 2025; Gou et al., 2024) are trained to operate with an expressive action and observation space using primitive operations like clicks and keystrokes, without the need for external accessibility elements. However, there is no SWE-specific environment for evaluating and further training these agents. `PwP` fills this gap by providing a unified, expressive IDE platform that challenges computer-use agents with realistic and diverse SWE tasks.

**Expressive Agent Environments**   Prior work on expressive agent environments has predominantly targeted the web domain (Koh et al., 2024; Deng et al., 2023), entire operating systems (Xie et al., 2024; Bonatti et al., 2024; Rawles et al., 2023), or other general scenarios (Xu et al., 2024). Some of these environments, such as OSWorld (Xie et al., 2024), feature general action and observation spaces similar to ours. However, although these benchmarks are capable of expressing a wide range of tasks, they do not focus on the unique challenges inherent to software engineering within an IDE. For example, while OSWorld offers a broad set of tasks, it is not specifically designed for SWE, resulting in increased computational overhead. Software engineering is a diverse and important domain that merits its own dedicated environment.

Additionally, we design `PwP` so that existing tool-based software engineering agents can be readily incorporated into our framework. Specifically, we modify the source code of the IDE to open up API calls that let us test current tool-based agents. Furthermore, `PwP-Bench` is tailored specifically for multimodal SWE tasks within an IDE, encompassing activities such as pull-request handling, debugging, and image-based code generation across multiple programming languages. We also observe that existing agents built for generic UI control often struggle in the `PwP` environment, as they must interact with a richer set of tools and achieve precise visual grounding within a complex interface containing a large number of interactive elements. We further distinguish `PwP` from other environments in Table 3.

## D   EXPERIMENTAL SETUP AND IMPLEMENTATION DETAILS

### D.1   AGENT DESIGN

In addition to the details mentioned in Section 5, we provide more implementation details in this section. First, the exact version numbers used for different API models are: gpt-4o-2024-11-20, gpt-4o-mini-2024-07-18, claude-3-5-sonnet-20241022, gemini-1.5-flash-preview-001, gemini-1.5-pro-preview-001, claude-3-7-sonnet-20250219, claude-sonnet-4-20250514. For the three Claude models, we use the computer-use variants by passing the 'computer-use' beta flag in API calls. For open-weights models, we run inference on 8 L40s using vLLM. We use temperature=0.3 consistently across models. For our main experiments, the number of iterations is set to 20 because: a.) for most tasks, 20 iterations is enough to complete the task, b.) increasing the number to more than 20 would increase the computational cost, and since some models didn't support caching at the time of running the experiments, the cost grows quadratically, c.) we ran experiments with 250 steps on Claude-Sonnet-4.0 on SWE-bench related datasets (see Appendix F); however, we found no difference in trends.

### D.2   MINI-SWE-AGENT

For mini-swe-agent, we use Claude-4.0 Sonnet. We use the same code as the official source code (SWE-agent, 2024), except that we modify it for multimodal tasks so that the agent receives required images as input in its prompt.

### D.3   OPENHANDS

We evaluate OpenHands, a strong baseline that uses SWE-specific prompts and has access to a variety of tools. Specifically, we use the `CodeActAgent` configuration with the following tools enabled: command line execution (`CmdRunAction`), IPython interactive shell (`IPythonRunCellAction`), and file operations (`FileReadAction`, `FileWriteAction`) and browser tool (`BrowserAction`). The agent also uses `AgentThinkAction` for reasoning and `AgentFinishAction` to conclude tasks. The agent uses Claude-Sonnet-4.0 with a temperature of 0.3.

## E   RESULTS

Table 8 presents comprehensive results for all agent designs across 15 datasets in `PwP-Bench`. Note, that non-trivial performance of mini-swe-agent-text on design2code and chartmimic is because

of noisy metrics, since if there is even some similarity between the actual image and the generated one (eg, chart type is accidentally the same), the metric returns a non-trivial score.

### E.1 COMPARISON WITH BEST REPORTED SPECIALIZED SWE AGENTS

In this section, we compare computer-use agents with the best reported specialized SWE agents scores on individual datasets. In particular, for each dataset, we use 3 different strategies to identify the best reported scores:

- **Citations**: For each dataset, we manually go through the citations and find the most relevant works and look for reported scores.
- **Official Leaderboard**: For some datasets, such as SWE-Bench, we use the official leaderboard to find the best reported scores.
- **Web-Search Agents**: We further prompt ChatGPT-5 thinking to find the latest and highest reported scores on each of the datasets. We then manually verify the results based on the links provided.

For each dataset, we follow all three strategies and take the highest reported score. Typically these results are achieved using specialized approaches including finetuned models, custom tool interfaces, specific pipelines, prompts, inference scaling, and verifiers. Therefore, it is important to note that direct comparisons on individual datasets may not provide a complete picture. Further, since our evaluations are done on 20 examples from the whole dataset, the results may not be directly comparable. Further, while we make our best effort to include the latest publicly available results, there may be still be discrepancies. Finally, the search was conducted on 22nd September 2025, and future numbers may change.

We now list the best reported scores for each dataset:

- **HumanEval**: QualityFlow (Hu et al., 2025) achieves 98.8% performance using Claude-3.5-Sonnet.
- **SWE-Bench**: Highest scores (75.2%) are achieved by a method named TRAE agent (Team et al., 2025), with best reported performance with Claude-4-Sonnet as base model as 74.6%.
- **SWE-Bench-Multilingual**: Highest score publicly reported is 43% (Yang et al., 2025) using Claude-3.7-Sonnet and Swe-agent framework (Yang et al., 2024a).
- **ResQ**: Highest score publicly reported is 58% (LaBash et al., 2024) using Claude-3.5-Sonnet in the official dataset report.
- **SWT-Bench**: Highest score publicly reported is 63.3% (Cassano et al., 2024) using GPT-4o in the official dataset report.
- **Design2Code**: Highest score publicly reported is 90.2% (Si et al., 2024a) using Claude-3.5-Sonnet in the official dataset report.
- **Chartmimic**: Highest score publicly reported is 86.46% using GPT-4o and METAL method (Li et al., 2025). Further they use inference scaling with n=5.
- **Intercode-CTF**: The publicly reported state of the art number is 72% using SWE-Agent-Enigma (Abramovich et al., 2024). This is much smaller than the numbers reported by our computer-use agent evaluation, which reaches 100% with the same Claude-3.5-Sonnet model. This is surprising, since the method employed numerous specialized tools for static analysis, dynamic analysis, and networking, and we confirmed that the improvement is statistically significant (p-value = 0.014, McNemar's test).
- **BIRD:** The best reported score is 76.14% (Shkapenyuk et al., 2025) as per the numbers reported in offciail leaderboard.
- **SWE-Bench-Multimodal**: The best reported score is 35.98% using scaffolding over O3, and 34.33% when using OpenHands-Versa (Soni et al., 2025) with Claude-4-Sonnet.

Overall, the results are often much higher than the numbers achieved by computer-use agents, even with access file and bash APIs. Overall, the discussion points out that at present specialized software-engineering agents still perform better, and built scaffolding around computer-use agents might also be helpful.

Table 8: Performance Evaluation of Different Models Across Task Categories. Leged: HE: HumanEval, SB: SWEBench, SJ: Swebench-Multilingual, RQ: ResQ, CI: CaniteEdit, ST: SWTBench, DC: Design2Code, CM: ChartMimic, DS: DSBench, SM: Swebench-MM, IC: Intercode-CTF, BD: Bird SQL, MC: Minictx, VS: VSCode, GS: General-SWE Tasks.

| Model | Code Generation & Editing (n = 6) | | | | | | Multimodal (n = 4) Code Generation | | | | Domain-Specific (n = 3) Code Generation | | | No-Code (n = 2) SWE Tasks | | Overall |
|---|---|---|---|---|---|---|---|---|---|---|---|---|---|---|---|---|
| | HE | SB | SJ | RQ | CI | ST | DC | CM | DS | SM | IC | BD | MC | VS | GS | Avg |
| *Computer-Use Agents* | | | | | | | | | | | | | | | | |
| Gemini-Flash | 0.0% | 0.0% | 0.0% | 0.0% | 0.0% | 0.0% | 15.2% | 2.0% | 0.0% | 0.0% | 0.0% | 0.0% | 0.0% | 5.0% | 0.0% | 1.1% |
| GPT-4o-mini | 0.0% | 0.0% | 0.0% | 0.0% | 5.0% | 0.0% | 14.8% | 0.0% | 0.0% | 0.0% | 0.0% | 0.0% | 0.0% | 5.0% | 0.0% | 1.7% |
| Qwen2.5-VL-72B | 0.0% | 0.0% | 0.0% | 0.0% | 0.0% | 0.0% | 17.1% | 0.0% | 0.0% | 0.0% | 0.0% | 0.0% | 0.0% | 10.0% | 0.0% | 1.8% |
| GUI-Owl-32B | 0.0% | 0.0% | 0.0% | 0.0% | 0.0% | 0% | 0% | 0% | 0% | 0% | 0.0% | 0.0% | 0.0% | 30.0% | 15.0% | 3.0% |
| Qwen3-VL-30B-A3B | 0.0% | 0.0% | 0.0% | 0.0% | 5.0% | 0.0% | 22.1% | 3.7% | 0% | 10.0% | 0.0% | 0.0% | 0.0% | 55.0% | 20.0% | 7.7% |
| Gemini-Pro | 10.0% | 0.0% | 0.0% | 0.0% | 5.0% | 0.0% | 14.5% | 8.1% | 0.0% | 0.0% | 0.0% | 0.0% | 0.0% | 15.0% | 0.0% | 3.5% |
| GPT-4o | 5% | 0.0% | 0.0% | 0.0% | 0.0% | 0.0% | 48.7% | 0.7% | 0.0% | 0.0% | 5.0% | 0.0% | 0.0% | 20.0% | 0.0% | 5.3% |
| Claude-Sonnet-3.5 | 20.0% | 0.0% | 0.0% | 15.0% | 25.0% | 4.2% | 18.1% | 0.0% | 5.0% | 10.0% | 15.0% | 0.0% | 0.0% | 35.0% | 10.0% | 10.5% |
| Claude-Sonnet-3.7 | 15.0% | 15.0% | 0.0% | 20.0% | 20.0% | 0.9% | 51.4% | 47.6% | 0.0% | 15.0% | 25.0% | 0.0% | 0.0% | 50.0% | 5.0% | 17.7% |
| Claude-Sonnet-4.0 | 16.7±5.8% | 5.0±5.0% | 5.0±0.0% | 21.7±2.9% | 20.0±0.0% | 17.3±3.8% | 60.9±2.2% | 68.3±4.4% | 10.0±0.0% | 10.0±0.0% | 20.0±0.0% | 0.0±0.0% | 0.0±0.0% | 56.7±7.6% | 23.3±2.9% | 22.3±0.5% |
| *Computer-Use Agents with File/Bash APIs* | | | | | | | | | | | | | | | | |
| Gemini-Flash | 0.0% | 5% | 5% | 15% | 15% | 17.1% | 19.9% | 13.5% | 3.2% | 10% | 25% | 0% | 0% | 5.0% | 0.0% | 8.9% |
| GPT-4o-mini | 60% | 10% | 5% | 20% | 30% | 16.7% | 41.3% | 5.5% | 8.4% | 15% | 40% | 5% | 0% | 10.0% | 0.0% | 17.8% |
| Qwen2.5-VL-72B | 10.0% | 5.0% | 0.0% | 25.0% | 25.0% | 17.1% | 34.1% | 13.1% | 0.0% | 0.0% | 5.0% | 15.0% | 0.0% | 15.0% | 0.0% | 11.0% |
| Gemini-Pro | 85% | 10% | 10% | 15% | 40.0% | 20.2% | 25.6% | 24.7% | 1.6% | 15% | 5% | 5% | 0% | 10% | 15.0% | 18.8% |
| GPT-4o | 85% | 25% | 10% | 30% | 50% | 17.0% | 70.2% | 65.5% | 11.9% | **20%** | 70% | 10% | 5% | 10.0% | 10.0% | 32.6% |
| Claude-Sonnet-3.5 | 95% | 25% | 10% | 55% | 65% | 37.4% | 83.4% | 71.2% | 55.7% | 10% | **100%** | 15% | 15% | 35.0% | 10.0% | 45.5% |
| Claude-Sonnet-3.7 | 90% | 25% | 15% | **65%** | **75%** | 41.4% | 79.2% | **81.2%** | **59.4%** | 15% | **100%** | 15% | 25% | 40% | 15.0% | 49.4% |
| Claude-Sonnet-4.0 | **100.0±0.0%** | **28.3±2.9%** | **25.0±0.0%** | 60.0±5.0% | 61.7±2.9% | **46.0±4.1%** | 87.1±2.1% | 77.4±2.1% | 53.3±5.3% | 13.3±2.9% | 96.7±5.8% | **18.3±2.9%** | 16.7±2.9% | 48.3±2.9% | **28.3±5.8%** | **50.7±0.2%** |
| *Software Engineering Agents* | | | | | | | | | | | | | | | | |
| mini-swe-agent | **100.0%** | 25.0% | 20.0% | 55.0% | 65.0% | 31.4% | **88.1%** | 80.2% | 57.9% | 15.0% | 90.0% | 10.0% | 20.0% | 55.0% | 20.0% | 48.8% |
| mini-swe-agent-text | **100.0%** | 25.0% | 20.0% | 55.0% | 65.0% | 31.4% | 40.4% | 8.1% | 50.1% | 10.0% | 90.0% | 10.0% | 20.0% | 55.0% | 20.0% | 40.0% |
| OpenHands | 95% | 25% | 20.0% | 55% | 70% | 37.4% | 85.2% | 68.9% | 38.9% | 10.0% | 85% | 15% | 30% | 30% | **20.0%** | 45.7% |

Table 9: **Tools available in different environments.** The table shows the various tools provided by different environments for assisted analysis. Common tools like file manipulation and bash operations are shared across environments, while specialized tools cater to specific tasks like web design and chart replication.

| Category | Tool | Description |
|---|---|---|
| Common Tools | bash | Perform bash operations |
| | file_edit | Perform file manipulation operations |
| SWEBench | search_repository | Search the repository for a string in the entire repository |
| | file_name_search | Search for a file by its name |
| | view_structure | View the structure of the current directory |
| Design2Code | view_html_preview | Get a preview of the index.html page as rendered in the browser |
| | view_original_image | Get a screenshot of the html image for replication |
| | zoom_in | Zoom in on the current rendered html page |
| | zoom_out | Zoom out on the current rendered html page |
| ChartMimic | view_python_preview | Get a preview of the graph generated by python file |
| | view_original_image | Get a screenshot of the graph for replication |
| BIRD | test_sql | Test a SQL query against the database |
| | get_relevant_schemas | Get relevant descriptions of the relevant database tables |

# F ADDITIONAL RESULTS

**Visual Grounding Errors.** In Section 5.1, we show that current agents struggle in visual grounding, despite some of these models being specifically trained for visual interfaces. To quantify the extent, we manually analyzed 20 random trajectories of two best performing agents: GPT-4o and Claude-3.5-Sonnet. In particular, we quantify the number of trajectories where the model had at least one visual grounding error, where a visual grounding error is defined as any of the following: (1) incorrect click, (2) incorrect interpretation of the current state, or (3) interacting with the wrong element. Surprisingly, we find that 20% of the trajectories of Claude-Sonnet-4.0, 35% for Claude-3.5-Sonnet, and 95% of the trajectories of GPT-4o contained at least one visual grounding error, indicating significant scope for improving these models for complex visual interfaces such as those demanded by PwP.

**Training models to use IDE tools better would improve performance.** In Section 5.1, we demonstrate that models can achieve superior performance when effectively utilizing IDE tools. In particular, Table 8 shows the performance of assisted agents (averaged across 3 models: GPT-4o, Gemini-1.5-Pro, and Claude-3.5-Sonnet), highlighting an average gain of up to 13.3%.

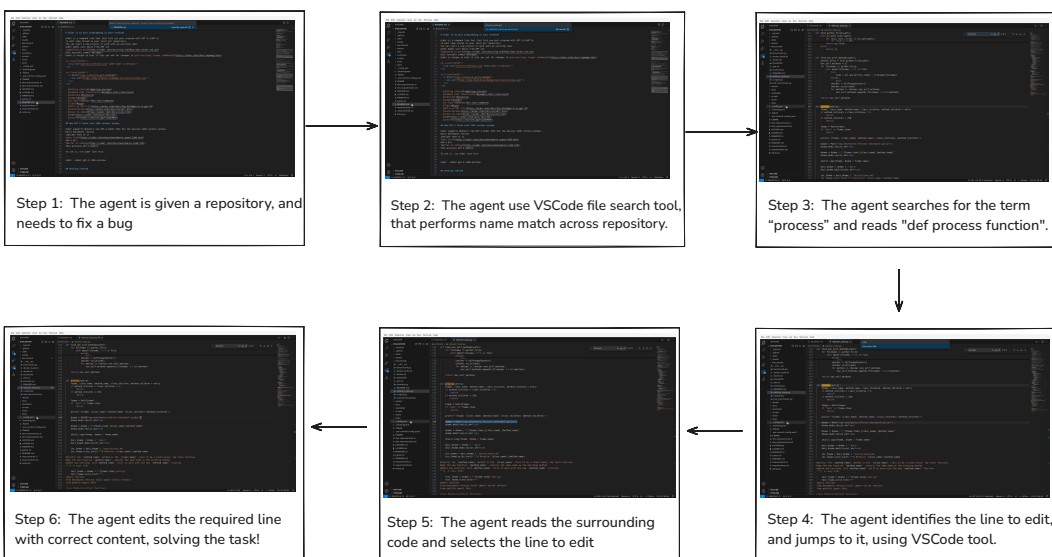

Figure 11: Example of the Claude Computer-Use agent successfully using multiple IDE tools to complete a repository level code-editing task.

However, our analysis reveals two primary limitations in current models' tool usage: (1) poor visual grounding and inability to handle complex tool interfaces, and (2) failure to prioritize IDE tool-based solutions over manual approaches.

To evaluate the second limitation specifically, we developed refactoring tasks within our 'General-SWE' dataset. These tasks require agents to rename symbols across a project repository—an operation that cannot be reliably accomplished through simple search-and-replace due to potential naming conflicts and contextual variations. The IDE provides a robust solution through its rename feature, which leverages the complete AST to ensure accurate symbol renaming across the codebase. This operation requires only pressing F2 on a symbol and entering the new name. In our evaluation, the Claude agent initially achieved 25% accuracy across four tasks when given no tool guidance. However, when explicitly prompted with "You can utilize the rename feature in VSCode to perform this task," its accuracy improved to 75%.

We observed similar patterns across other tasks designed to evaluate tool usage. For instance, tasks that could be efficiently solved using the debugger showed limited success. While agents could sometimes set breakpoints, their poor visual grounding prevented them from effectively interpreting the debugging interface—particularly in understanding the current execution state and paused line location. These findings suggest significant potential for improving agent performance through better training on IDE tool utilization.

**Successful Use of Tools** We further show a couple of examples of successful tool use in Figure 2 3. However, we do note that while the agent is able to use the IDE tool through UI interaction, it still may not be able to make optimal use of it as shown in Figure 17.

**Agents Fail to Edit Files.** File editing is a basic capability required in most SWE tasks. However, we find that the deficiencies in visual grounding significantly impact the file editing capabilities of current agents that use basic actions (clicking and typing). For example, even when provided with cursor location information in textual form, these models struggle to interpret such data amid complex UI elements. Models fine-tuned for UI interactions still commit basic editing errors—such as incorrect indentation and text misplacement—and are unable to recover from these errors (see Appendix for examples). We speculate these limitations could stem from two factors: (i) model overfitting to user interfaces in their training domains, or (ii) the increased complexity of the `PwP` IDE interface, which contains substantially more interactable elements than typical web or OS environments. Addressing these limitations represents an important direction for future work. Although direct file access via tool operations is available, UI-based editing confers unique advantages for

Table 10: Performance Evaluation of Different Agents on 250 steps on SWE-Bench related tasks.

| Model | SWE-Bench | SWE-Bench-Multimodal | SWE-Bench-Multilingual | Average |
|---|---|---|---|---|
| **Computer-Use Agent** | 10.0% | **30.0%** | 15.0% | 18.3% |
| **CUA w/ File/Bash Tools** | **60.0%** | **30.0%** | **40.0%** | **43.3%** |
| **mini-swe-agent** | **60.0%** | **30.0%** | 35.0% | 41.7% |

tasks such as editing Jupyter notebooks, comparing changes, or modifying specific sections of large files. These results underscore two limitations: (i) current VLMs are challenged by complex UI interactions beyond simple web/OS interfaces (Xie et al., 2024; Koh et al., 2024), and (ii) the inability to effectively perform UI-based editing prevents agents from leveraging valuable IDE features that could have improved their performance.

**Agents Are Incapable of Recovering from Errors.** Next, we find that current agents show limited error recovery capabilities. When an action fails to execute correctly, models tend to persistently repeat the same failed action without exploring alternatives. Similarly, if an agent selects an incorrect action, it continues along an erroneous solution path without recognizing or correcting the mistake. In an experiment designed to probe this behavior, we deliberately suppressed one of the model's (Gemini-1.5-Pro) actions. Despite the environment's screenshot clearly showing an unchanged state, the models proceeded with their planned action sequence as though the suppressed action had succeeded. This behavior suggests a heavy reliance on memorized action sequences rather than dynamic responses to visual feedback, resulting in exponentially increasing errors and poor performance. However, when we repeated the experiment with Claude-Sonnet-4.0, we tested 5 such scenarios, and found only in one case, the agent ignored the screenshot, potentially highlighting that computer-use agents are improving over time.

**Performance on Long Horizon Tasks.** In our main experiments, we had capped the maximum number of agent steps to 20, owing to high cost associated with each of the models. However, certain datasets, such as SWE-Bench, typically require much larger number of steps for agent to complete the task. In this section, we therefore evaluate 3 agents based on Claude-Sonnet 4.0, with 250 steps on 3 relevant datasets: SWE-Bench, SWE-Bench-Multimodal, SWE-Bench-Multilingual. The results are shown in Table 10. We note, that almost all agents show consistent improvement in performance with higher number of steps. However, overall tends remain consistent with 20 steps: Computer-Use Agents with File/Bash APIs show 43.3% performance, and mini-swe-agent shows 41.7% performance, and pure computer-use agents show 18.3% performance.

**Robustness and Ablations** To address potential robustness concerns, we re-ran the experiments for the best-performing computer-use agent (with and without bash/file APIs) over multiple trials. We find the variance in performance on `PwP-Bench`-Lite is very low (1.x%), and does not affect any of the conclusions of the work.

We also investigated the importance of visual modality by running mini-swe-agent without providing any images on the multimodal task category. Overall, the performance drops significantly, demonstrating the importance of the visual modality. However, the effect varies by task: performance on SWE-Bench-MM sees a non-significant difference, whereas performance on tasks like ChartMimic drops to near zero.

**IDE Feature Usage Analysis** We analyzed the distribution of IDE feature usage across all trajectories of the Claude-Sonnet-4.0 Computer-Use Agent. Specifically, we prompted Claude-Sonnet-4.0 with complete trajectory (including textual steps and each step's screenshot), to generate what feature the model used at each step. We then aggregate this information across all tasks, and report the results in Figure 12. Importantly, the results just show when the agent attempted to use a tool, and not whether it succeeded or not. As shown in Figure 12, the agent primarily relies on basic editing and navigation features. Specifically, Text/Code Editing accounts for 26.2% of interactions, followed by File Explorer usage (16.8%), Terminal interaction (12.5%), and Global Search (10.6%). These four categories alone comprise over 65% of all IDE interactions. In contrast, advanced features see significantly lower usage: Debug/Run capabilities are used in only 1.8% of interactions, and Code Intelligence features (such as go-to-definition) account for just 2.8%. Further, manual inspection

shows that while the agent is attempting to use these advanced features, it fails very commonly. This analysis reinforces our finding that while current agents can perform basic IDE operations, they struggle to leverage the full depth of specialized software engineering tools available in the environment.

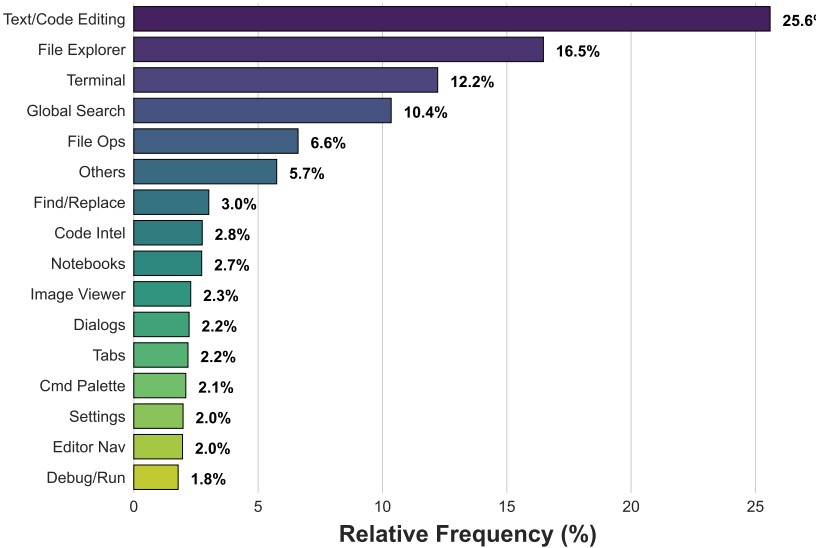

Figure 12: Distribution of VS Code Feature Usage by Claude-Sonnet-4.0 Agent. The agent predominantly uses basic editing and navigation tools, with limited usage of advanced features like debugging or code intelligence.

## G  QUALITATIVE ANALYSIS

In this section, we consider both positive and negative examples of agent grounding and ability to interact with the complete IDE interface in `PwP`.

## H  DISCUSSION

**Computational Overhead of Running PwP**  While `PwP` provides a much more general interface for software engineering agents, a natural question is what computational overhead it introduces. The added computational requirements primarily come from: (1) capturing screenshots using the xdotool library, (2) running the IDE, (3) maintaining a VNC server, and (4) processing video and audio streams via ffmpeg. Importantly, only components (1) and (2) are essential for all agents, as video and audio processing are only necessary when agents must interpret visual or auditory cues—a universal requirement for any environment supporting these modalities. The VNC server is used solely for debugging or pair programming scenarios and can be disabled when not needed. The xdotool commands consume negligible CPU resources (¡¡ 1%) and minimal memory. While VSCode does increase memory and CPU utilization, the latency overhead remains limited, and the computational cost is substantially lower than running the large-scale computer-use models that power the agents. In summary, despite its comprehensive feature set, the computational overhead of `PwP` is minimal, with the primary computational demand stemming from the computer-use models themselves rather than the environment.

**Why use IDE over simple Bash Agent?**  While computer-use agents perform worse than even simple API based SWE agents, intuitively there still remains a lot of value in utilizing a general interface such as IDE, for software engineering. The reason being modern IDEs, have been developed over multiple years of effort, and provide several advantages that are not possible with say bash interface. While, theoretically it may still be possible to create equivalent tools, it would take similar tremendous effort, to develop them again for agents, with less reliability.

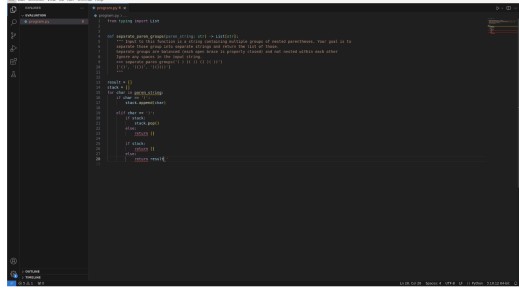

Figure 13: **Example of Agent Missing Visual Error Indicators** The agent fails to recognize linter error indicators (wavy underlines).

Figure 14: **Example of Agent's Inability to Perform File Editing** The agent incorrectly positions new content in the file editor.

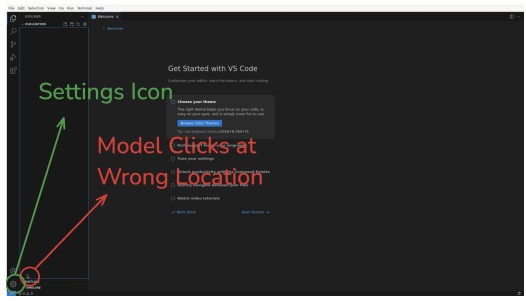

Figure 15: **Example of wrong mouse click by Claude-Computer Use Agent** The agent attempted to click Settings icon but clicked at the wrong location.

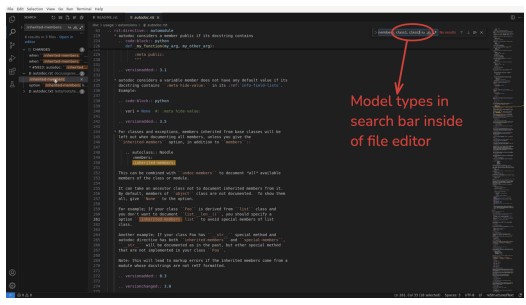

Figure 16: **Example of Agent Misidentifying Active Panel** The agent fails to recognize the active editor panel, incorrectly typing into the search bar (red arrow) instead of the file editor.

To give few examples of myriads advantages of IDEs:

- **Interactive Debugging Capabilities**
  - IDEs provide rich, stateful debugging interfaces that allow AI agents to set breakpoints, inspect variables, and evaluate expressions dynamically
  - Unlike CLI debuggers (GDB, LLDB, pdb), IDE debuggers maintain visual context and state, making it easier for AI agents to track program flow and debug complex scenarios
  - The visual representation of stack traces and variable states is more structured and machine-parseable compared to text-based CLI output

- **Intelligent Code Refactoring**
  - IDEs maintain a complete Abstract Syntax Tree (AST) of the project, enabling accurate symbol renaming and code restructuring across multiple files
  - AI agents can leverage IDE's semantic understanding to perform complex refactoring operations with higher confidence
  - Unlike text-based search-and-replace in Bash, IDE refactoring tools understand code context and prevent accidental modifications to unrelated symbols

- **Test Management and Coverage Analysis**
  - IDEs provide structured APIs for test discovery, execution, and result analysis
  - AI agents can efficiently track test coverage through visual indicators and programmatic interfaces
  - Real-time test feedback and coverage data is more readily accessible compared to parsing CLI test runner output

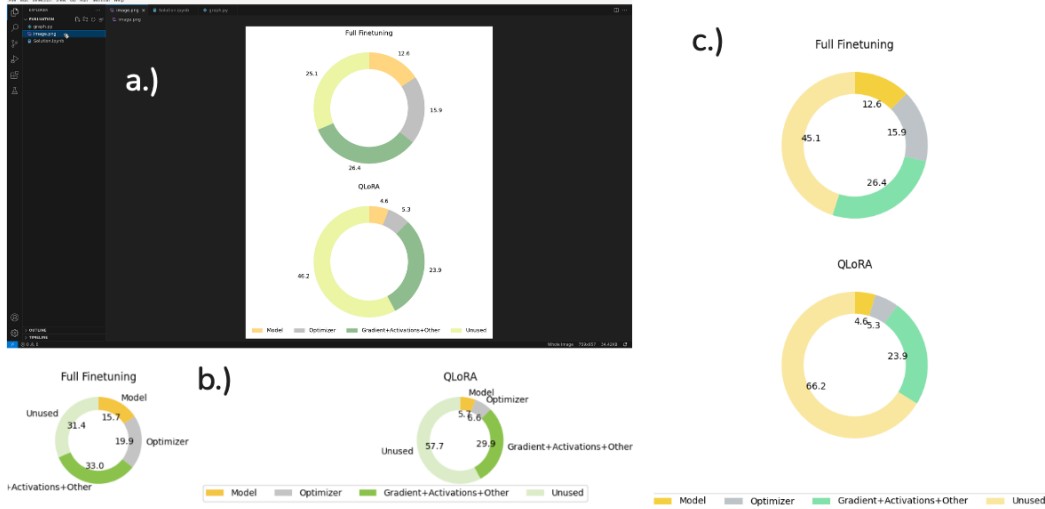

Figure 17: Performance comparison of GPT-4 agent in Computer-Use and Assisted settings on the ChartMimic dataset. a) Image as seen by the Computer-Use agent. b) Replication in Computer-Use setting. c) Replication in Assisted setting. The Assisted agent demonstrates superior performance despite seeing the same image but in different context and state.

- **Performance Profiling and Analysis**
  - IDE profilers offer structured data about CPU usage, memory allocation, and runtime behavior
  - Visual representations of performance metrics (flame graphs, memory usage) are easier for AI agents to analyze systematically
  - Profiling data is available through APIs rather than requiring parsing of complex text-based output

- **Code Indexing and Semantic Search**
  - IDEs maintain comprehensive code indexes that enable fast, context-aware code search and navigation
  - AI agents can leverage these indexes for more accurate code understanding and modification
  - Unlike grep or find, IDE search capabilities understand code structure and can filter based on semantic properties

- **Extension Integration and Automation**
  - IDE extensions can be programmatically controlled through APIs, allowing AI agents to leverage additional tools seamlessly
  - Extensions can provide structured data and interfaces that are more reliable for automation compared to parsing CLI tool output
  - Configuration and coordination of multiple tools can be managed through unified IDE interfaces rather than managing separate CLI tools

## I  PROMPTS

In this section, we provide the prompts used for the agents in our evaluation.

### I.1  COMPUTER-USE AGENT PROMPTS

```
1  system_message = """You are an autonomous intelligent agent tasked with interacting with a code IDE (e.g.,
       VSCode). You will be given tasks to accomplish within the IDE. These tasks will be accomplished through
       the use of specific actions you can issue.
```

```
2
3   Here's the information you'll have:
4
5   - **The user's objective**: This is the task you're trying to complete.
6   - **The current IDE screenshot**: This is a screenshot of the IDE, with each clickable element assigned a
            unique numerical ID. Each bounding box and its respective ID shares the same color.
7   - **The observation**, which lists the IDs of all clickable elements on the current IDE screen with their text
            content if any, in the format '[id] [element type] [text content]'. The element type could be a button,
            link, textbox, etc. For example, '[123] [button] ["Run"]' means there's a button with id 123 and text
            content "Run" on the current IDE screen.
8   - **Delta Image**: The difference between the current image and the previous image, highlighting the changes
            that have occurred. You can use this information to figure the action executed by you had the intended
            effect or not. Additionally, this serves purpose of clearly showing the content that you may want to
            focus on.
9   - **The cursor position**: Information about the current cursor position, provided as a DOM element in both
            text and image formats.
10
11  The actions you can perform fall into several categories:
12
13  ---
14  You can use the computer_control tool to issue these actions. example, you can call the computer_control tool
            and pass arguments as 'xdotool type "hello world"' to type "hello world" at the current cursor position.
15
16  **Keyboard Actions:**
17
18  - 'xdotool type "[content]"': Type the specified content at the current cursor position.
19  - 'xdotool key [key_combination]': Simulate pressing a key or combination of keys (e.g., 'xdotool key "ctrl+s
            "' to save a file).
20
21  **Mouse Actions:**
22
23  - 'xdotool mousemove [id] click [click_code]': Move the mouse to the element with the specified id and click
            on it.
24  - 'xdotool mousemove [x] [y] click [click_code]': Move the mouse to the coordinates (x, y) on the screen and
            click.
25  - 'xdotool mousemove [id] click --repeat 2  1': Double-click on the element with the specified id.
26  - 'xdotool mousemove [id] click 5': Scroll down on the element with the specified id.
27  ---
28
29  **IDE Navigation Actions:**
30
31  - **Interacting with IDE Tools**: You can use any tools inside the IDE, such as file explorer search, go to
            definition, etc., by performing the appropriate keyboard or mouse actions.
32
33  ---
34
35  **Completion Action:**
36
37  - 'execution_done': Issue this command when you believe the task is complete. Do not generate anything after
            this action.
38
39  ---
40
41  To be successful, it is very important to follow these rules:
42
43  1. **Start with a Plan on how to achieve the objective**:
44    - Begin the task by creating a plan on how you will achieve the objective. Think about the steps you need to
            take, the tools you can use, and the actions you need to perform. This will help you stay organized and
            focused throughout the task.
45
46  2. **Start every step with an Image Description**:
47    - Begin every step by describing the provided IDE screenshot.
48    - Enclose your description within '<image description>' tags.
49    ```
50    <image description>
51    <!-- Your description here -->
52    </image description>
53    ```
54
55  2. **Think Before Acting**:
56    - Analyze the screenshot and plan your next action carefully.
57
58  3. **Issue Only Valid Actions**:
59    - Only perform actions that are valid given the current observation.
60
61
62  5. **Completion**:
63    - Use 'execution_done' when you think you have achieved the objective.
64
65
66  6. **Cursor Positioning**:
67    - Before editing any file or field, make sure where the cursor is. Clear things if you have already written
            something, and do not want it anymore. You can also move cursor to right location use vscode utility by
            sending key ctrl+g, typing line number, press Return, then move to the right column using arrow keys.
68    - If unsure, use keyboard shortcuts or mouse actions to place the cursor appropriately before typing.
69   - Despite being sure, you might still make a mistake. Review screenshots and text information after each
            action to ensure correctness.
70
71  7. **Precision in Actions**:
72    - Be precise when performing mouse actions.
73    - Prefer keyboard actions over mouse actions whenever possible.
74
```

```
75  8. **Utilize Available Tools**:
76     - Leverage any functionalities available within the IDE to accomplish the task.
77
78
79  9. **Other Tips**:
80     - **Use UI:** Use UI when possible instead of editing files.
81     - **Review the text on Screen**: Previous experiments with you show, that you often confuse what is shown
          in the image. Make sure you use the text information provided to cross verify what you are seeing in the
          image.
82     - **Learn from Mistakes**: If a an action or step of actions didn't get the intended result, think of
          different strategy in order to achieve the goal.
83     - **Keyboard Shortcuts**: Use keyboard shortcuts whenever possible to increase efficiency. For instance, in
          order to open settings, use "xdotool key ctrl+," instead of clicking on the settings icon.
84     - **What is on Screen**: If you do not see something in a menu/setting that you were planning to use, look
          for appropriate search bar, and type relevant queries to find the option you are looking for.
85     - **Clear the Editor/Input Field**: If you are planning to type something in an editor or input field, make
          sure to clear the existing content before typing the new content. For instance, you can use "xdoool key
          ctrl+a BackSpace" to clear the content.
86     - **Location:** Do not automatically assume you are at the right location before typing. For instance, if
          you want to search something, make sure your cursor is in the right input field. If nothing gets typed,
          despite the command being correct, you are supposed to find the right input field and click on it and
          then type again.
87     - **VSCode Shortcutts:** VScode shortcuts are not necessarily same as xdotool commands. For example in
          order to execute ctrl+k ctrl+o, you will have to use two commands: `xdotool key ctrl+k` followed by `
          xdotool key ctrl+o`.
88  ---
89
90  **Remember**: Your actions should methodically guide the IDE towards accomplishing the required task, using
        precise and atomic commands. Prioritize keyboard interactions over mouse actions to enhance efficiency.
91
92  <IMPORTANT FILE EDITING>
93  Keep in mind these tips while editing files:
94  - To jump to a particular line number, you can use `ctrl+g` followed by `line number` (and optionally column
        number, eg: 11:12) and then press `Return`.
95  - If you execute a type command, however, file does not change, it can likely mean, the focus is not on the
        file. Make sure to move your mouse to the file and click on it, to ensure the file is focused.
96  - While typing make sure that correct indentation is being used.
97  """
```

## I.2 MINI-SWE-AGENT PROMPTS

```
1   system_template: |
2     You are a helpful assistant that can interact with a computer.
3
4     Your response must contain exactly ONE bash code block with ONE command (or commands connected with && or
        ||).
5     Include a THOUGHT section before your command where you explain your reasoning process.
6     Format your response as shown in <format_example>.
7
8     <format_example>
9     Your reasoning and analysis here. Explain why you want to perform the action.
10
11    ```bash
12    your_command_here
13    ```
14    </format_example>
15
16    Failure to follow these rules will cause your response to be rejected.
17  instance_template: |
18    Please solve this issue: {{task}}
19
20    You can execute bash commands and edit files to implement the necessary changes.
21
22    ## Recommended Workflow
23
24    This workflows should be done step-by-step so that you can iterate on your changes and any possible
        problems.
25
26    1. Analyze the codebase by finding and reading relevant files
27    2. Create a script to reproduce the issue
28    3. Edit the source code to resolve the issue
29    4. Verify your fix works by running your script again
30    5. Test edge cases to ensure your fix is robust
31    6. Submit your changes and finish your work by issuing the following command: `echo
        COMPLETE_TASK_AND_SUBMIT_FINAL_OUTPUT`.
32       Do not combine it with any other command. <important>After this command, you cannot continue working on
        this task.</important>
33
34    ## Important Rules
35
36    1. Every response must contain exactly one action
37    2. The action must be enclosed in triple backticks
38    3. Directory or environment variable changes are not persistent. Every action is executed in a new
        subshell.
39       However, you can prefix any action with `MY_ENV_VAR=MY_VALUE cd /path/to/working/dir && ...` or write/
        load environment variables from files
40
41
42    ## Formatting your response
```

```
43
44    Here is an example of a correct response:
45
46    <example_response>
47    THOUGHT: I need to understand the structure of the repository first. Let me check what files are in the
       current directory to get a better understanding of the codebase.
48
49    ```bash
50    ls -la
51    ```
52    </example_response>
53
54    ## Useful command examples
55
56    ### Create a new file:
57
58    ```bash
59    cat <<'EOF' > newfile.py
60    import numpy as np
61    hello = "world"
62    print(hello)
63    EOF
64    ```
65
66    ### Edit files with sed:
67
68    ```bash
69    # Replace all occurrences
70    sed -i 's/old_string/new_string/g' filename.py
71
72    # Replace only first occurrence
73    sed -i 's/old_string/new_string/' filename.py
74
75    # Replace first occurrence on line 1
76    sed -i '1s/old_string/new_string/' filename.py
77
78    # Replace all occurrences in lines 1-10
79    sed -i '1,10s/old_string/new_string/g' filename.py
80    ```
81
82    ### View file content:
83
84    ```bash
85    # View specific lines with numbers
86    nl -ba filename.py | sed -n '10,20p'
87    ```
88
89    ### Any other command you want to run
90
91    ```bash
92    anything
93    ```
```

