# OpenReview forum: "Programming with Pixels: Can Computer-Use Agents do Software Engineering?"
_ICLR.cc/2026/Conference — ICLR 2026 Poster_

### Official Review · Reviewer_KWZS · 2025-11-02

**Soundness:** 3
**Presentation:** 3
**Contribution:** 3
**Rating:** 8
**Confidence:** 4

**Summary:**

The paper benchmarks multimodal LMs on solving software engineering tasks using an IDE.
This PwP benchmark is built from many existing benchmarks, plus 20 tasks that have been added.
Overall, the paper presents results based on a random sample of 300 task instances (though the full benchmark contains more than 5000).
It clearly demonstrates that when only given computer use controls, all models significantly underperform on almost all areas of tasks when compared to models that are given simple APIs.  The paper also makes some observations about how performance could be improved on the benchmark.

**Strengths:**

Significance: Computer use is a very relevant skill and the constructed benchmark is very challenging (with a current best score of 23% for computer-use agents) while staying close to real-world use cases. Especially if LMs were to be evaluated on the full set of > 5000 task instances, this would show a very fine-grained picture of LM skills. Because so many different benchmarks are combined, the data obtained is rich and could be analyzed even further. Overall, I expect this benchmark to become relevant as a computer-use agent benchmark especially as easier benchmarks get saturated.

Originality: The benchmark is mostly constructed from existing benchmarks, but using them to evaluate computer use is new.

Quality: The paper is solidly executed and evaluating on tasks from so many different existing benchmarks, as well as running computer use agents to scale is not easy. The quantitative and qualitative observations on computer use agent behavior are interesting.

Clarity: Well written and easy to understand.

**Weaknesses:**

I mostly had some questions about the composition of the benchmark as well as the score, I've put them all in the questions section.

Minor presentation details:

* p.6 official name seems to be mini-swe-agent (with dashes) (also appears completely without dashes in tab1). It might also be good to call your version mini-swe-agent multimodal or similar or clarify in tab 1 (when I first looked at tab 1, I thought this was the off-the-shelves mini-swe-agent, i.e., without any multimodal support built in)
* Tab 1: There are two "best numbers" in bold in column 3
* Tab 1 caption should point out that these are the results on the lite split of the dataset. It would also be helpful to put `n=` for the different evaluated categories

**Questions:**

* It would be interesting to have a baseline completely without multimodal tools, for example by running an unmodified mini-swe-agent. This would be an obvious comparison for code generation & editing, but some multimodal problems can also be solved without access to screenshotting etc. (and if not this would show us the extent to which the benchmarks _require_ multimodality to be solved). For example many instances SWE-bench multimodal can definitely be solved without having the agent use screenshots for verification etc. (though including the image in the initial task instance might be required for some instances)
* Since the pwp benchmark mixes so many very different benchmarks, understanding how the categories are made up and how the lite split is built might be main-text-worthy. Maybe just mentioning on p. 4+5 what the percentage of each of the benchmarks is in the corresponding category? Or even just some `n=` number of instances in tab 1 to understand how many instances are evaluated.
* If I understand the appendix correctly, and the paper samples 20 instances for the lite split from each task source, then this also leads to very different numbers of tasks for the different categories (e.g., General SWE tasks only has 20 instances in total), so these instances might have a disproportionate sway over the overall avg if you take the unweighted average. It will also significantly increase the statistical uncertainty on the scores.
* Could you show the prompts that you use for the agent in the appdx? One through-line of the paper is the comparison of task-specific software engineering agents (like mini-swe-agent) and general purpose agents. However, mini-swe-agent seems to be pretty general, too, since it doesn't have any specific tools for software engineering (other than that the prompt mentions a few strategies). So unless your prompt is much more general, I'm not sure if this generalized vs specialized narrative really is supported.
* How can mini-swe-agent (even with screenshot support) score in some tasks of the "General SWE task" category? The appdendix for example gives "installing of vscode extensions" as an example. Can this be done from the bash?
* Since there are only 20 tasks that are added and because they have a strong influence on the overall score that is reported, it might be worth putting a summary of each of them in the appendix.

---

> ### Author Response · Authors · 2025-11-21
>
> We thank the reviewer for their thoughtful and positive review. We appreciate your recognition of our benchmark's significance, originality, and quality, and we are glad you found no major weaknesses.
>
>
> We answer your questions below:
>
> ---
>
> > Minor presentation details:
>
> Thank you for these suggestions. We have incorporated all of them in the revised paper.
>
> ---
>
> > It would be interesting to have a baseline completely without multimodal tools, for example by running an unmodified mini-swe-agent. This would be an obvious comparison for code generation & editing, but some multimodal problems can also be solved without access to screenshotting etc. (and if not this would show us the extent to which the benchmarks require multimodality to be solved). For example many instances SWE-bench multimodal can definitely be solved without having the agent use screenshots for verification etc. (though including the image in the initial task instance might be required for some instances)
>
> We have now run mini-swe-agent without providing any images on the multimodal task category. Overall, the performance drops by an average ~33% (on multimodal category), demonstrating the importance of the visual modality. However, the effect varies by task: performance on SWE-Bench-MM sees a non-significant difference, whereas performance on tasks like ChartMimic drops to near-trivial scores. We have added these results to Appendix F.
>
> ---
>
> > Since the pwp benchmark mixes so many very different benchmarks, understanding how the categories are made up and how the lite split is built might be main-text-worthy. Maybe just mentioning on p. 4+5 what the percentage of each of the benchmarks is in the corresponding category? Or even just some n= number of instances in tab 1 to understand how many instances are evaluated.
>
> Thank you for this suggestion. We have added the number of task (n=) for each category to Table 1 and mentioned it in the main text.
>
> > these instances might have a disproportionate sway over the overall avg if you take the unweighted average. It will also significantly increase the statistical uncertainty on the scores.
>
> There is no disproportionate sway because the "Overall Avg" in Table 1 is a macro-average across all 15 tasks, not the four high-level categories. Each task contributes equally. Regarding statistical uncertainty, we have re-run experiments over multiple trials and found the standard deviation to be very low (0.7%), confirming the robustness of our results.
>
> ---
>
> > Could you show the prompts that you use for the agent in the appdx? One through-line of the paper is the comparison of task-specific software engineering agents (like mini-swe-agent) and general purpose agents. However, mini-swe-agent seems to be pretty general, too, since it doesn't have any specific tools for software engineering (other than that the prompt mentions a few strategies). So unless your prompt is much more general, I'm not sure if this generalized vs specialized narrative really is supported.
>
> We have now included all prompts in Appendix I. Our distinction is based on the agent's action space. A generalist CUA uses a universal, low-level action space (pixels, keyboard, mouse) and can theoretically perform any task on a computer. On the other hand, a specialized SWE agent, even a general one like mini-swe-agent, uses an API that constrains its functionality. For example, it’s not clear how to have mini-swe-agent interact with a GUI by clicking. To further clarify this distinction, we have also added the OpenHands baseline, which uses even more SWE-specific tools and prompts (detailed in Appendix D).
>
> ---
>
> > How can mini-swe-agent (even with screenshot support) score in some tasks of the "General SWE task" category? The appdendix for example gives "installing of vscode extensions" as an example. Can this be done from the bash?
>
> Yes, it turns out that some of the tasks can be completed using vscode cli, eg. models can simply install themes/extensions etc.
>
> ---
>
> > Since there are only 20 tasks that are added and because they have a strong influence on the overall score that is reported, it might be worth putting a summary of each of them in the appendix.
>
> Thanks for the suggestion, we have included the information in the appendix in the revised paper.
>
> ---
>
> Thank you again for your constructive feedback and positive assessment. We believe these clarifications and additions have further strengthened the paper.

---

### Official Review · Reviewer_Syoz · 2025-11-02

**Soundness:** 2
**Presentation:** 2
**Contribution:** 3
**Rating:** 4
**Confidence:** 3

**Summary:**

This paper introduces a new environment and benchmark to test whether general-purpose computer-use agents (CUAs) can perform complex software engineering (SWE) tasks. The authors develop Programming with Pixels (PwP), a VSCode-based environment where agents interact visually via clicks, typing, and screen observation to complete tasks such as code editing, debugging, and UI design. They also propose PwP-Bench, a 15-task benchmark covering multiple languages and modalities, to systematically assess agent performance. Through extensive evaluation, the study finds that purely visual CUAs perform far below specialized coding agents, but when granted limited text-based APIs (file editing and bash), their performance nearly doubles, approaching specialized agent levels. The analysis highlights poor visual grounding and limited tool usage as key weaknesses, while emphasizing the rapid progress and potential of CUAs for future generalist software engineering applications.

**Strengths:**

- This paper introduces Programming with Pixels (PwP), the first comprehensive IDE-based environment that allows computer-use agents to perform real software engineering tasks through visual interaction, offering a realistic and general testbed. They proposes PwP-Bench, a well-designed and diverse benchmark spanning 15 tasks across multiple programming languages, modalities, and software engineering domains—covering code, UI, data science, and IDE configuration.

- It provides clear empirical insights showing how adding minimal text-based APIs dramatically boosts agent performance, helping isolate the impact of visual grounding and tool-use limitations.

- It establishes software engineering as a meaningful and economically significant domain for evaluating generalist computer-use agents, setting the stage for future research toward specialist-level generalist systems.

**Weaknesses:**

- There is limited discussion on the cost, efficiency, or scalability of running PwP which are important factors in order to serve as a community standard benchmark.

- Although PwP mimics a realistic IDE, the experiments do not explore longer, multi-step development workflows or collaborative settings common in real-world software engineering.

- The evaluation metrics are primarily accuracy-based and may not fully capture partial progress, reasoning quality, or the richness of multi-step interaction trajectories.

**Questions:**

Since file and bash APIs drastically boost performance, do the authors believe that future progress should prioritize improving visual grounding or integrating more structured textual interfaces? and how might that affect the generalist nature of computer-use agents?

---

> ### Author Response · Authors · 2025-11-21
>
> We thank the reviewer for their thoughtful review and appreciation of PwP as the first comprehensive IDE-based environment, well-designed PwP-Bench spanning diverse tasks, clear empirical insights, and establishing software engineering as a meaningful domain for evaluating generalist computer-use agents.
>
> We are happy to address your questions and suggestions below:
>
> ---
>
> > There is limited discussion on the cost, efficiency, or scalability of running PwP which are important factors in order to serve as a community standard benchmark.
>
> Our PwP environment is designed to be efficient, scalable, and cost-effective. We clarify this in Appendix H, noting that the environment's computational overhead is negligible. The system is quite scalable: "Each container is isolated, preventing interference between experiments, ensuring parallel evaluation" (Lines 191-192). We have discussed that the running cost of the environment is negligible (Appendix H), using limited memory and CPU. Further evaluations are scalable since “Each container is isolated, preventing interference between experiments, ensuring parallel evaluation” (Lines 191-192). Further, we specifically provide PwP-Bench-Lite (300 instances) for cost-efficient evaluation. PwP is also easy to run, since environment setup requires only a single pip command (Line 198).
>
> ---
>
> >  Although PwP mimics a realistic IDE, the experiments do not explore longer, multi-step development workflows or collaborative settings common in real-world software engineering.
>
> Our work focuses on understanding the *current* limitations of state-of-the-art CUAs on established, foundational SWE tasks for AI agents. Our results show that today's agents struggle even with these self-contained, single-developer problems (e.g., zero successful uses of the debugger) showing limited performance on aggregate (22.9\% accuracy) . Therefore, although investigating more complex, long-horizon workflows is an interesting future work, we think that it would provide limited additional insights on the limitations of current agents. PwP provides the necessary testbed to solve these fundamental challenges first. At the same time, as agent capabilities improve, our environment is explicitly designed to evolve; "adding new benchmarks is as simple as modifying configuration files" (Line 184).
>
> ---
>
> > The evaluation metrics are primarily accuracy-based and may not fully capture partial progress, reasoning quality, or the richness of multi-step interaction trajectories.
>
> We agree that pass/fail metrics are coarse, a common characteristic of current execution-based SWE benchmarks.  However, our work provides a much deeper analysis than the final score alone. We conduct qualitative and quantitative trajectory-level analysis to understand *why* agents fail, identifying key mechanistic issues related to visual grounding, inability to use ide features, inability to edit files, etc (Section 5.1). We quantify visual grounding errors (20\% and 95\% of trajectories for GPT-4o and Claude-4.0 respectively, Appendix F) and, in the revised paper, add a quantitative breakdown of IDE feature usage. As a result, we believe our analysis provides several actionable insights for future research that are not evident from accuracy alone.
>
> > Since file and bash APIs drastically boost performance, do the authors believe that future progress should prioritize improving visual grounding or integrating more structured textual interfaces?
>
> Our analysis (Section 5.1) identifies **both** as critical: (1) visual grounding failures prevent basic IDE interaction (Figures 4, 5, 12, 14), and (2) inability to leverage IDE tools limits performance even when visual grounding succeeds. We therefore believe improving on both as important future directions.
>
> ---
>
> Thank you again for your review. We hope these responses address your concerns. If issues remain, we look forward to discussing them further. Alternatively, if you feel your concerns are addressed, we would appreciate if you could reflect that in your evaluation.

---

### Official Review · Reviewer_5ZwW · 2025-11-09

**Soundness:** 2
**Presentation:** 3
**Contribution:** 2
**Rating:** 2
**Confidence:** 3

**Summary:**

This paper introduces PwP, an environment for evaluating the software engineering (SWE) capabilities of computer-use agents (CUAs). It also presents PwP-Bench, a benchmark covering 15 task types, including newly proposed datasets for VSCode-use and general SWE tasks. Experimental results show that current CUAs perform poorly on SWE-related tasks. Only when a bash tool is added do they achieve performance comparable to a purely text-based mini-swe-agent that relies solely on bash operations.

**Strengths:**

- Evaluating CUAs on more complex tasks to comprehensively assess their capabilities is an important and valuable direction.

- The paper aggregates a wide range of existing datasets and adds two new SWE-related task sets, resulting in a large and diverse benchmark.

**Weaknesses:**

- The motivation is somewhat confusing. General-purpose CUAs are primarily designed for everyday computer-use tasks, such as web navigation or basic UI operations. While CUAs **can** perform SWE tasks, that doesn’t necessarily mean they **should** be expected to do well on SWE. This feels similar to asking *whether an image generation model can produce good videos*—theoretically possible, but not aligned with its intended design or primary evaluation purpose.

- I also find it difficult to identify the essential difference between PwP-Bench and SWE-Bench (-MM). If the goal is to test SWE performance, then a simple bash environment with read, write, and execute (-rwx) capabilities is sufficient (which is also the design strategy of Mini-SWE-Agent). Notably, code is inherently *textual*. Even web front-end layouts are generated from *textual style configs*. If one wants to expose the model to rendered visual outputs, webpage-based inputs (as used in SWE-Bench-MM) already achieve this. Could the authors clarify what is the core different between using webpages as vision input a  using IDE screenshots? Furthermore, everything visible in an IDE can already be accessed in text form, without introducing noise or visual ambiguity. Thus, if the goal is to evaluate SWE ability, feeding noisy visual input to the agent seems counterproductive. PwP-Bench assumes that the agent must read from IDE images,  which introduces an artificial constraint that inherently weakens any agent’s performance on SWE tasks.

- Moreover, the authors seem aware of the limitation of solely using visual input for SWE: They add a bash tool to CUAs, and indeed, Claude-Sonnet-4.0 with bash achieves the best performance—nearly identical to Claude-Sonnet-4.0 Mini-SWE-Agent. Again, it is a very, very small agents with only **100 lines** of code and **only one simple bash tool**. This strongly suggests that SWE tasks are simply not well-suited for general-purpose CUAs. Even a minimal agent—an LLM with just a bash tool—can match or outperform them.
This further reinforces the view that CUAs are not suited for solving SWE tasks, thereby undermining the claimed contribution of this work.

I agree that CUAs should be tested on more complex tasks, but SWE may not be the right choice. Since CUAs’ key advantage lies in their visual interaction capabilities, it might make more sense to evaluate them on tasks that genuinely depend on visual understanding, rather than on code-centric problems that are mostly text-based.

**Questions:**

- What is the fundamental difference between using IDE images as input in PwP-Bench and using webpages as input in SWE-Bench-MM?
- Is a CUA with bash tools still a general-purpose CUA? Does it become a SWE-specialized CUA or even a vision-enabled SWE agent?What exactly are we evaluating here?

---

> ### Author Response · Authors · 2025-11-21
>
> We thank the reviewer for their thoughtful review and for recognizing that evaluating CUAs on complex tasks is an important direction, and for appreciating the scale and diversity of our benchmark.
>
> We address your concerns below:
>
> ---
>
> > The motivation is somewhat confusing. General-purpose CUAs are primarily designed for everyday computer-use tasks, such as web navigation or basic UI operations.
>
> We respectfully disagree with this perspective, so we would like to clarify the motivation for our work here. Computer-use agents are in principle very general: they *could* automate any task that involves interacting with software on a computer. This includes many tasks beyond everyday computer-use tasks. While it is true that so far the community has primarily evaluated computer-use agents on web navigation or basic ui operations, those tasks do not constitute the set of tasks that a computer-use agent can be used for. Indeed, expanding the set of tasks that computer-use agents are evaluated on is exactly the motivation of our work.
> To draw an analogy, language models a few years ago were typically evaluated on traditional NLP tasks, while (to take one example), program synthesis was done using non-neural specialized models and algorithms. At the time, one could have argued that language models were not designed for program synthesis, let alone problem solving, scientific discovery, or the wealth of tasks that they are now evaluated on. Indeed, today’s language models are now evaluated across a wide portfolio of tasks to test both breadth and depth.
>
> In a similar vein, we argue that since in principle a computer-use agent could perform any task achievable on a computer, computer-use agents should be evaluated on a wide variety of computer tasks. This includes specialized tasks like software engineering, which our work aims to do. As we discuss in the introduction, software engineering is particularly interesting because there are very strong baselines to beat; namely, the wealth of software-engineering agents being developed by the community. Going back to our analogy, we want to see how an agent that is in principle very general (analogous to a prompted language model) performs compared to an agent that is more specifically designed for the specialized task (analogous to a model specifically designed for machine translation). We hope this clarifies the motivation for our work.
>
> ---
>
> >  I also find it difficult to identify the essential difference between PwP-Bench and SWE-Bench (-MM). If the goal is to test SWE performance, then a simple bash environment with read, write, and execute (-rwx) capabilities is sufficient (which is also the design strategy of Mini-SWE-Agent). Notably, code is inherently textual. Even web front-end layouts are generated from textual style configs. If one wants to expose the model to rendered visual outputs, webpage-based inputs (as used in SWE-Bench-MM) already achieve this. Could the authors clarify what is the core different between using webpages as vision input a using IDE screenshots?
>
> Our work has a completely different motivation from SWE-Bench-MM: SWE-Bench-MM is not aimed at evaluating computer-use agents. For example, it is not clear what GUI the computer-use agent would be clicking on/interacting with in SWE-Bench-MM.
> In contrast, our goal is to **evaluate computer-use agents** on a set of specialized computer-use tasks. To recap, in principle a computer-use agent could feasibly perform any task that involves interacting with software on a computer, so the goal of our work is to frame software engineering as interacting with software on a computer, and to see to what extent computer-use agents are currently good or bad at it. We choose software engineering because it can be framed as a computer-use task (i.e., interacting with an IDE), it has economic importance, is challenging, and has numerous specialized agents serving as strong baselines (Lines 44-48).

---

> > ### Author Response · Authors · 2025-11-21
> >
> > > Furthermore, everything visible in an IDE can already be accessed in text form, without introducing noise or visual ambiguity. Thus, if the goal is to evaluate SWE ability, feeding noisy visual input to the agent seems counterproductive
> >
> > To reiterate, our goal is to evaluate computer-use agents. We believe that the potential difficulty of perceiving text from images is *exactly what we are aiming to test*, since it is a key functionality for a computer-use agent to have.   Furthermore, the IDE is particularly information-dense, which makes it particularly interesting for evaluating computer-use agents.Indeed, our analysis gives concrete evidence that future computer-use agents need to have a better ability to perceive and interact with such information-dense interfaces.
> > Furthermore, while everything visible in the IDE can be largely accessed in text form , such a representation may still be challenging for the agent to use. For example, the SoM agent in our evaluation uses HTML DoM information, but underperforms some of the vision-only agents.
> >
> > ---
> >
> > > Moreover, the authors seem aware of the limitation of solely using visual input for SWE: They add a bash tool to CUAs, and indeed, Claude-Sonnet-4.0 with bash achieves the best performance—nearly identical to Claude-Sonnet-4.0 Mini-SWE-Agent. Again, it is a very, very small agents with only 100 lines of code and only one simple bash tool. This strongly suggests that SWE tasks are simply not well-suited for general-purpose CUAs. Even a minimal agent—an LLM with just a bash tool—can match or outperform them. This further reinforces the view that CUAs are not suited for solving SWE tasks, thereby undermining the claimed contribution of this work.
> >
> > Our analysis indeed highlights that visual grounding is one of the key limitations of current computer-use agents. However, we never claim that current general purpose CUAs should or should not be used for SWE, and this is not our claimed contribution. Our contribution is evaluating CUAs on the specialized task of software-engineering, and highlighting weaknesses of current CUAs through our  benchmark.
> > Whether CUAs are suited for SWE is a topic of debate (eg, Anthropic showcased their computer-use agent release by releasing video of CUA using vscode [2]). Drawing an analogy to language models again, one could have argued a few years ago that language models were not suited for a variety of tasks that they now perform well on. However, our work does not take any position on this debate, and rather provides an environment and benchmark for evaluating computer-use agents.  We respectfully ask that our work is evaluated based on our stated contributions (Lines 96-103), rather than on the question of whether CUAs are suited for SWE.
> >
> > ---
> >
> > We thank the reviewer again for reviewing our paper. We look forward to addressing any concerns the reviewer may still have during the discussion period!

---

> ### Comment · Reviewer_5ZwW · 2025-11-21
>
> Thanks for author’s effort on clarification. I am sure that our main disagreement is not about the technical soundness, but rather about ***whether it is necessary to design a purely SWE-focused evaluation suite for CUA***. From this perspective, I want to clarify my view: *SWE can certainly be included in CUA evaluation, but it should not constitute the entirety of it.* A comprehensive CUA benchmark should span a diverse set of tasks, including (but not limited to) SWE itself. This also aligns with the authors’ claim that CUAs should be “evaluated across a wide portfolio of tasks to test both breadth and depth.”
>
> Given this, if the paper’s goal is to propose a **SWE-only** benchmark specifically for CUAs, I would raise an intuitive question: ***Why can’t existing SWE benchmarks be adapted for CUA evaluation to fill the gap of SWE coverage in current CUA benchmarks?*** Let me be more specifically: This connects to several points:
>
> 1. *CUA can use visual input, but it should also be able to choose using text-only input*. In my view, a capable CUA should know when to rely on the most suitable input modality. The authors argue that “the difficulty of perceiving text from images is *precisely what we aim to test.*” I agree this is a true challenge for VLM, but I think the goal of a CUA is to *solve problems*. Using the modality that naturally has a comparative advantage seems more intuitive, rather than forcing the agent to do something it is not good at.
>
>     I understand the authors’ goal is to *evaluate CUAs’ capability on SWE rather than claiming that CUA should be / should not be used for SWE.* I do not deny it. I am only speaking from the angle of the experimental results: I believe the prospects of this direction are not promising (of course this is my subjective opinion and does not objectively negate the work):
>
>     - **We can measure many methods A/B/C/D on tasks 1/2/3/4, but when we consider whether a task is *worth evaluating*, there should be some inspiring experimental evidence to justify it** (For example, we can evaluate whether a image generator a good video generator. But it does not drive any timely and insightful conclusion for the improvement in this area.). Thus, unfortunately, I feel discouraged by the outcome that CUA + shell achieves **nearly the same** performance as **mini-swe-agent**, which is only around **100 lines of code**. This brings me back to the core question I raised earlier:
>
>         **“Is a CUA with bash tools still a general-purpose CUA? Does it become a SWE-specialized CUA or even a vision-enabled SWE agent? What exactly are we evaluating here?”**
>
>         If a CUA must be pushed close to a (very very lightweight) SWE-specialized agent in order to complete SWE tasks, then is it realistic to evaluate the SWE capability of CUAs in the first place
>
>         Even if we measure that one CUA is better than another CUA in SWE, ***what conclusion can be derived**?*  Users still would not use CUAs for SWE since it is not good for such task, and CUA’s developers might not enhance CUA’s SWE ability since it is not their major domain.
>
>         This situation never happens with more diverse but simpler tasks, such as web navigation, or slightly more complex tasks (e.g., “Every Saturday night open YouTube to watch a livestream, but don’t open it if there is a cancellation email”). These are tasks that **only** a CUA can do, and that CUAs excel at. If some CUAs perform well on such tasks, that metric directly guides users’ or developer’s future decesions.
>
> 2. The authors say “it is not clear what GUI the computer-use agent would be clicking on/interacting with in SWE-Bench-MM,” but the evaluation result of SWE tasks has nothing to do with “clicking/interacting.” It depends solely on whether the final tests pass. This does not seem to be a strong rebuttal.
>
>     Furthermore, whether a CUA can solve SWE tasks is not even related to whether it uses an IDE. For example, claude code can perform efficient SWE entirely through shell interaction. Even if this process lacks visual input, the webpage provided by SWE-Bench-MM compensates for that.
>
>     Considering the authors’ explanation:
>
>     “We choose software engineering because it can be framed as a computer-use task (i.e., interacting with an IDE), it has economic importance, is challenging, and has numerous specialized agents serving as strong baselines,”
>
>     I believe that directly using existing shell-based agents as baselines + webpage as the visual input is obviously a more economical choice. This combination is actually the *dual* of the authors’ strategy of CUA + shell:
>
>     - one has visual ability but no shell.
>     - one has shell but no visual ability

---

> > ### Comment · Reviewer_5ZwW · 2025-11-21
> >
> > 3. The author claim that
> >
> >     > *“Indeed, our analysis gives concrete evidence that future computer-use agents need to have a better ability to perceive and interact with such information-dense interfaces. Furthermore, while everything visible in the IDE can be largely accessed in text form, such a representation may still be challenging for the agent to use. For example, the SoM agent in our evaluation uses HTML DoM information, but underperforms some of the vision-only agents.”*
> >     >
> >
> >     I am not sure what specific results this part is referring to. I might have missed key information. Could the authors point out the exact location?
> >
> >
> > Although I have many ***subjective*** concerns about this work, I still agree with the authors’ original objective: designing more diverse and more complex tasks.
> >
> > However, again, I believe **pure SWE is not the right direction**.
> >
> > I do not want to influence other reviewers’ views (who, in my opinion, are very positive about this work). My comments are solely to provide one perspective. Though I view this work negatively, I will *not* argue that it must be rejected.
> >
> > If most people buy into the idea, then perhaps this paper suits their scenarios, but not mine.
> >
> > Therefore, I adjust my rating to 4. I do not mind whether this paper can be accept or not (there are plenty of paper accepted every year). I just want to explain my view: a more diverse, essential, and hard benchmark (maybe with *more complicated orchestration tasks for daily life*) is needed for CUA evaluation.

---

> > > ### Author Response · Authors · 2025-12-02
> > >
> > > We are glad that the reviewer **does not find any technical weakness** in our work, and explicitly states they "will not argue that it must be rejected."
> > > We appreciate that the disagreement has *converged to the subjective question* of whether SWE-specific evaluation for CUAs is valuable. We are also glad that our work has led to discussion on these multiple perspectives. While we cannot resolve all subjective differences in this thread, we address three specific points:
> > >
> > > > I believe pure SWE is not the right direction... a more diverse, essential, and hard benchmark is needed for CUA evaluation
> > >
> > > We agree that CUAs should be evaluated on diverse tasks: we never claimed SWE should be the only domain. PwP-Bench is intended to complement existing CUA benchmarks (OSWorld, WebArena, etc.) by adding a **more challenging**, **economically important** domain. Just as language models today are evaluated on a wide set of tasks including code, math, and creative writing, our contribution is establishing and facilitating SWE as one such important domain for CUAs.
> > >
> > > > I feel discouraged by the outcome that CUA + shell achieves nearly the same performance as mini-swe-agent
> > >
> > > We understand this concern, but we see this as evidence of rapid progress rather than discouragement. Figure 7 shows pure CUAs improved from 10.5\% to 22.9\% over just 7 months (2.2x improvement), and Table 1 shows the gap between pure CUAs and CUAs with APIs has decreased from 35.0\% to 27.8\%. Ideally, we would have loved for CUAs to already leverage IDE tools effectively: unfortunately they currently struggle (as discussed in the results section), but they are clearly improving, and our benchmark provides a way to track this progress. At the same time, the result establishes PwP as a challenging benchmark for CUAs.
> > >
> > > > I am not sure what specific results this part is referring to. I might have missed key information. Could the authors point out the exact location?
> > >
> > > First, the evidence that IDE interfaces are particularly challenging appears in Section 5.1 (Lines 416-421), where we state "the deficiencies come from the IDE being particularly information-dense." Specific failure modes are illustrated in Figures 4, 5, 13, 14, and 15.
> > > Second, the SoM comparison can be found in Table 1. Models that lack native UI grounding and therefore depend on SoM/DOM annotations (e.g., GPT-4o: 5.3\%, Gemini-Pro: 3.5\%) underperform models with native visual understanding (Claude-Sonnet-4.0: 22.3\%), despite all receiving the same SoM annotations. We also discuss this in Lines 406-408: "Surprisingly, Set-of-Marks did not resolve these issues; agents would instead select incorrect elements."
> > >
> > > ---
> > >
> > > We thank the reviewer for raising their score and for the constructive discussion, which concluded with no technical concerns.

---

### Official Review · Reviewer_pzk4 · 2025-11-10

**Soundness:** 3
**Presentation:** 3
**Contribution:** 3
**Rating:** 6
**Confidence:** 4

**Summary:**

Following are the key components of the research article, although there are issues with some of the sections that will be listed later.

Problem Statement:
- The problem statement is clearly articulated in the Introduction (Page 1-2):
- Primary question: "can general-purpose computer-use agents match specialized agents in complex domains like software engineering?"
- Motivation: Current evaluations of computer-use agents focus on simple tasks, leaving unclear whether they can handle sophisticated tasks like software engineering.

State of the Art:
- Related work is covered in Section 2 (Page 2-3):
- Comprehensive coverage of multimodal and computer-use agents
- Software engineering agents literature
- Comparison with existing environments and benchmarks

Method Explanation
- The methods are explained in Section 3 (Pages 3-5):
- Programming with Pixels (PwP) environment as a POMDP
- PwP-Bench benchmark with 15 tasks across multiple categories
- Evaluation methodology clearly described

Results Presentation
- Results are presented in Section 5 (Pages 5-11):
- Table 1 shows comprehensive performance evaluation
- Clear comparison between different agent types
- Detailed analysis of limitations and capabilities

Claims vs Results
- The abstract makes specific claims that are supported by results:
- Claim: "pure visual interaction performs significantly worse" -> Supported by 22.9% vs 48.8% performance
- Claim: "file editing and bash operations dramatically improve performance" -> Supported by 50.7% accuracy with APIs
- Claim: "clear room for future improvements" -> Supported by visual grounding analysis and improvement potential demonstrated


The authors systematically address whether computer-use agents can handle software engineering by creating a comprehensive IDE environment that captures real software engineering complexity. Then they evaluate across diverse tasks spanning 15 different scenarios, which provides fair comparison with specialized agents through the same environment. Authors state that no prior comprehensive environment existed for this domain. While individual components exist (computer-use agents, SWE benchmarks), their combination is novel. Table-3 mentions which areas were covered by other works as compared to this work.

**Strengths:**

The PwP environment addresses a critical gap in evaluating computer-use agents for software engineering tasks. By providing a realistic IDE interface through VSCode, the environment enables comprehensive testing of agent capabilities while maintaining the expressiveness needed for diverse software engineering activities. The extensible design, with features like checkpointing, multimodal support, and easy benchmark addition, positions it as a lasting contribution to the research community.

The study maintains fidelity to established benchmarks by using programmatic verifiers and metric calculations that match the original dataset specifications. This approach ensures meaningful comparisons with prior work and establishes a reliable baseline for future research, while the 15-task PwP-Bench provides comprehensive coverage of software engineering activities.

The combination of detailed performance tables with concrete examples of agent behavior (such as visual grounding errors, successful tool usage, and failure modes) provides a balanced perspective. This multi-faceted analysis helps explain why certain performance patterns emerge, making the results actionable for both understanding current limitations and guiding future improvements in computer-use agent design.

**Weaknesses:**

The following are key issues identified in the work

- The reliance on PwP-Bench-Lite (300 instances) due to computational constraints creates potential sampling bias, potentially missing important edge cases that would emerge in full-scale evaluation. The 20-step limitation, while computationally necessary, may artificially constrain performance on genuinely complex software engineering tasks, though the supplementary 250-step experiments provide partial validation.

- The heavy focus on Claude-based agents (3.5, 3.7, 4.0) introduces anthropocentric bias, as these models may share architectural similarities that don't represent the broader landscape of computer-use agents. Additionally, evaluating only VSCode limits generalizability across the diverse IDE ecosystem used by developers worldwide.

- The inconsistent file editing capabilities across models and the limited investigation into API-driven improvements suggest insufficient mechanistic understanding of why certain interventions succeed. The study documents performance differences but lacks deeper analysis of the underlying factors driving these disparities, limiting its prescriptive value for future agent development.

**Questions:**

The following recommendations address critical gaps in statistical rigor and model diversity that would significantly strengthen the study's scientific validity:

- The current study's minimal statistical testing (only one p-value reported) represents a fundamental weakness for a comprehensive benchmarking effort. How the p-value is derived is also missing. Implementing proper statistical methodology with confidence intervals, effect sizes, and multiple comparison corrections would transform the findings from descriptive observations to statistically significant conclusions. Power analysis would ensure the 300-instance sample size is adequate for detecting meaningful differences, while systematic statistical testing across all 15 datasets would provide robust evidence for claimed performance gaps.

- The absence of mixture-of-experts models represents a significant oversight, as these architectures are increasingly important in the LLM landscape and may exhibit different performance characteristics on computer-use tasks. Expanding beyond Claude-centric evaluation to include these diverse model architectures would provide crucial insights into whether performance differences stem from fundamental model design choices rather than implementation details. This broader coverage would make the benchmark more representative of the current AI landscape and more valuable for future research direction.


The following are a few minor issues identified in the article
- Page 5: "evluate" should be "evaluate"
- Page 11: "intutively" should be "intuitively"
- Page 13: "potetntially" should be "potentially"
- Page 21 (and some other places): "os" should be "OS"

---

> ### Author Response · Authors · 2025-11-21
>
> We thank the reviewer for their thoughtful review and acknowledging that our PwP environment addresses a critical gap in evaluating CUAs, comprehensive testing, extensible design, meaningful comparison with baselines, and our comprehensive analysis.
> We are happy to address your questions and suggestions below:
>
> ---
>
> > The reliance on PwP-Bench-Lite (300 instances) due to computational constraints creates potential sampling bias, potentially missing important edge cases that would emerge in full-scale evaluation.
>
> To address potential sampling bias, we randomly sampled 20 tasks from each of the 15 benchmarks (Lines 863-864).
> Further, creating PwP-Lite is our strength as it enables efficient evaluation while the full 5400 instances remain available if needed. Moreover, the size of PwP-Lite is standard in both CUA literature (eg, 369 instances in OSWorld) and SWE benchmarks (eg, 500 instances in SWE-Bench-Verified).
>
> ---
>
> > The 20-step limitation, while computationally necessary, may artificially constrain performance on genuinely complex software engineering tasks, though the supplementary 250-step experiments provide partial validation.
>
> As you noted, we conducted 250-step experiments (Table 8, Appendix F) showing "overall trends remain consistent with 20 steps" (Line 1240). Therefore, we don't suspect that the 20-step limit artificially constrains our conclusions. We also noticed that, roughly speaking, most tasks require fewer than 20 steps (for instance, design2code, humaneval, resq, etc). That said, if you suspect that a particular benchmark requires a larger step size, we would be happy to evaluate it with a larger step size during the discussion period.
>
> ---
>
> > The heavy focus on Claude-based agents (3.5, 3.7, 4.0) introduces anthropocentric bias, as these models may share architectural similarities that don't represent the broader landscape of computer-use agents.
>
> We respectfully disagree that our evaluation is Claude-centric: we evaluated **11 different models** from diverse families: GPT (4o, 4o-mini), Gemini (Flash, Pro), Qwen (2.5-VL-72B, GUI-Owl-32B, Qwen3-VL-30B-A3B), and Claude (3.5, 3.7, 4.0), including both open-weights and closed-source models. Parts of our analysis focused on Claude models since they consistently achieve state-of-the-art performance, allowing us to understand the current frontier of CUA capabilities. Nonetheless, our results and analysis spans across multiple different models.
>
> ---
>
> > The inconsistent file editing capabilities across models and the limited investigation into API-driven improvements suggest insufficient mechanistic understanding of why certain interventions succeed. The study documents performance differences but lacks deeper analysis of the underlying factors driving these disparities, limiting its prescriptive value for future agent development.
>
> Section 5.1 provides analysis that shows at least 3 major factors that underlie the poor performance of CUA agents: visual grounding failures (Figures 4, 5, 12, 14), file editing difficulties (Figure 13), and inability to use IDE features (Lines 421-428). We quantified that 20\% and 95\% of trajectories have visual grounding errors for GPT-4o and Claude-4.0 respectively (Appendix F). We have also added quantitative IDE feature usage analysis in the revised paper (Appendix F). We believe these analyses provide important directions of research for future work.
>
> ---
>
> > The current study's minimal statistical testing (only one p-value reported) represents a fundamental weakness for a comprehensive benchmarking effort ... systematic statistical testing across all 15 datasets would provide robust evidence for claimed performance gaps.
>
> We have re-run experiments for our best-performing CUA (both with and without file/bash apis) over multiple trials and found the standard deviation to be very low (0.7\%), thereby providing evidence for reported performance gaps, and adequacy of PwP-Bench-Lite's size.
>
> ---
>
> > The absence of mixture-of-experts models represents a significant oversight, as these architectures are increasingly important in the LLM landscape and may exhibit different performance characteristics on computer-use tasks.
>
> We have now included Qwen3-VL-30B-A3B, which uses a MoE architecture. Note that many closed-source models may already be using MoE internally. Further, our evaluation spans several architectures and providers (OpenAI, Anthropic, Google, Qwen). As a result, we respectfully disagree that our evaluation is Claude-centric, and would be happy to discuss it further with you here.
>
> ---
>
> Thank you again for your thoughtful review. We hope our responses and the additional experiment have addressed your concerns. If you think there are still some concerns, we look forward to addressing them as well! Alternatively, if you feel all your concerns are addressed, we would appreciate it if you could reflect that in your evaluation!

---

> > ### Comment · Reviewer_pzk4 · 2025-11-27
> >
> > Thank you for addressing all the issues pointed out. I am satisfied with the rebuttals.

---

### Official Review · Reviewer_xKNb · 2025-11-10

**Soundness:** 3
**Presentation:** 4
**Contribution:** 3
**Rating:** 6
**Confidence:** 3

**Summary:**

The paper introduces an environment to approach software engineering using a visual interface. The visual interface consists of screenshots, sometimes enhanced by the set-of-mark technique, which provides a list of elements in addition to the screenshot. They provide an evaluation benchmark consisting of 15 tasks with an OS and a VSCode-like IDE. The VSCode IDE is changed so that it supports the set-of-mark technique and gives the DOM tree.

They then conducted a comprehensive evaluation of different computer-use agents and found that they perform poorly on this benchmark using only visual inputs and actions like 'type' and 'click'. Then they added two functionalities that boost the performance of agents. They gave them text-based terminal access and text-based file editing tools. These two additional text-based abilities boosted the performance of agents. They also investigated and saw that on some tasks the agents mostly use the text-based interface, while on others they only use the visual interface.

The paper also investigates why pure visual agents fail, identifying two main issues. The first and primary weakness is poor visual grounding. The second issue is the agent's inability to leverage sophisticated IDE features, such as debuggers or profilers, even when these tools could simplify the task.

**Strengths:**

1. The provided benchmark is very valuable, and its container-based approach makes evaluations consistent.
2. The provided VSCode IDE that returns screenshots with set-of-marks and DOM is valuable and can be used in future research.
3. The evaluation of various computer-use agents is comprehensive.
4. The ablation studies are also comprehensive, and the results on how much the text-based interface vs. the visual interface is used are very interesting.

**Weaknesses:**

1. To measure the importance of the visual interface, it would be interesting to measure the performance when agents only have access to text-based tools (file edit, terminal, and task-specific tools). It would show the computer-use agents' ability in text-only settings.
2. Some tasks aren’t relevant to agentic software development. For example, changing the theme of the IDE is not that important, while similar IDE setting changes, like turning on auto-suggestion, could be helpful.
3. Some deeper insights into which IDE features agents are using would be very beneficial to add to the paper. For example, can the paper provide a number on whether the agent uses debugging tools, or if it ever did? The framework would benefit if these insights are generated after running tests, which would help those developing agents using this benchmark.

**Questions:**

1. The testing environment needs more clarification on its safeguards, as it has been seen that LLMs might simply remove tests if they have access to those files. Do they have access to them through bash, file editing, or the visual interface, in theory?

---

> ### Author Response · Authors · 2025-11-21
>
> We thank the reviewer for their thoughtful review and appreciation of the value of our benchmark, consistent and comprehensive evaluations, comprehensive ablations and potential of future research based of our environment.
>
> We are happy to address your questions and suggestions below:
>
> ---
>
> > To measure the importance of the visual interface, it would be interesting to measure the performance when agents only have access to text-based tools (file edit, terminal, and task-specific tools). It would show the computer-use agents' ability in text-only settings.
>
> Our baseline mini-sweagent provides exactly this comparison as it uses only text-based APIs. Further, we have just added an additional agent baseline, OpenHands, that uses SWE specific text tools such as FileEdit, TaskTracking, IPython, etc. Overall, the results highlight that despite using the same base models, computer-use agents underperform these text-only API models by a significant margin, primarily because of poor visual grounding and file-editing capabilities (Figure 4,5,12,14), highlighting significant scope of improvement in future work.
>
> ---
>
> > Some tasks aren’t relevant to agentic software development. For example, changing the theme of the IDE is not that important, while similar IDE setting changes, like turning on auto-suggestion, could be helpful.
>
> We kindly note that our proposed VSCode dataset does have tasks such as turning on auto-suggestions. It also includes some simple tasks, such as changing the theme. These tasks primarily test whether agents can navigate IDE menus and settings, which is a prerequisite for using advanced IDE features. Based on your feedback, we also now include a full list of tasks in Appendix B in the revised paper.
>
> ---
>
> > Some deeper insights into which IDE features agents are using would be very beneficial to add to the paper. For example, can the paper provide a number on whether the agent uses debugging tools, or if it ever did? The framework would benefit if these insights are generated after running tests, which would help those developing agents using this benchmark.
>
> We agree that adding this analysis would further strengthen the existing analysis. To recap, in our existing analysis in Section 5.1 ("Agents Struggle to Use Advanced IDE Functionality," Lines 421-428), we qualitatively note that agents successfully use basic tools like the file explorer and live preview (Figures 2 and 11), but fail to use advanced ones. Further, we observed zero successful uses of debuggers or profilers across all evaluated models, despite these tools being available (Section 5.1, Appendix E). To add to this analysis and to address your comment, in the revised paper, we have further quantified the IDE features used across all trajectories of Claude-Sonnet-4 CUA. The results indicate the agents primarily rely on basic file editing and navigation tools, however there are non-trivial instances of using debuggers and code intelligence tools, although failing to use them successfully. Full details are added in Figure 12, Appendix F.
>
> ---
>
> > it has been seen that LLMs might simply remove tests if they have access to those files. Do they have access to them through bash, file editing, or the visual interface, in theory?
>
> This is a great question. In short, we specifically designed PwP such that LLMs **do not have access to any tests.** In PwP, agents are run in a separate Docker environment and evaluations are done in a separate one. This separation makes it impossible for agents to remove or modify any tests.
>
> ---
>
> Thank you again for your thoughtful review. We hope our responses and the additional experiment have addressed your concerns. If you think there are still some concerns, we look forward to addressing them as well! Alternatively, if you feel all your concerns are addressed, we would appreciate it if you could reflect that in your evaluation!

---

> > ### Comment · Reviewer_xKNb · 2025-11-27
> >
> > Thank you for the rebuttal. You have addressed my concerns, and I have updated my score.

---

### Official Review · Reviewer_f9o7 · 2025-11-10

**Soundness:** 2
**Presentation:** 3
**Contribution:** 3
**Rating:** 6
**Confidence:** 3

**Summary:**

This paper presents Programming with Pixels (PwP), the first IDE-based environment and benchmark for evaluating computer-use agents (CUAs) on software-engineering tasks. PwP enables agents to control a VSCode IDE visually through keyboard and mouse, while PwP-Bench unifies 15 multi-language, multimodal SWE tasks. The authors systematically compare pure visual CUAs, CUAs with minimal text APIs, and specialized SWE agents. Results show that visual CUAs perform poorly (22.9%) but reach near-specialist performance (50.7%) when given file-edit and bash APIs. PwP thus establishes software engineering as a fair, comprehensive testbed for generalist computer-use agents.

**Strengths:**

Addresses an important intersection between two major agent paradigms — IDE-based (CUA) and API-based (SWE) — by providing a unified testbed for fair comparison.

Novel environment design: the first fully interactive VSCode-based setup enabling realistic software-engineering evaluation.

Comprehensive benchmark: integrates 15 diverse SWE tasks across modalities and languages.

Insightful analysis: identifies visual grounding and tool-usage limitations as key failure modes, offering concrete directions for improvement.

**Weaknesses:**

Only one specialized SWE baseline (Mini-SWEAgent) is included; broader baselines would strengthen comparisons.

Conceptual clarity: definitions of computer-use agent vs SWE agent and the rationale for “fair evaluation” (why expressiveness and access matter) could be better explained.

PwP-Bench-Lite covers many task types but has very few instances (20 per task), and the paper reports no repeated trials or variance, raising robustness concerns.

Technically, the benchmark adapts existing datasets rather than introducing new ones.

**Questions:**

see weakness

---

> ### Author Response · Authors · 2025-11-21
>
> Thank you for your thoughtful review and positive assessment of our work. We appreciate your recognition of our work being important at the intersection of CUA and SWE, the novelty of our environment, our comprehensive benchmark, and our insightful analysis.
>
> We are happy to address your questions and suggestions below:
>
> ---
>
> > Only one specialized SWE baseline (Mini-SWEAgent) is included; broader baselines would strengthen comparisons.
>
> Thank you for the suggestion. We have updated the paper with an additional specialized SWE agent: OpenHands. OpenHands is a strong baseline that uses SWE-specific prompts and has access to a variety of tools such as FileEdit, FileRead, FileWrite, CmdRun, Ipython, TaskTracking, Context Condensation, and Browser Usage. We find that this agent scores similarly to Mini-SWE-Agent, and these new results do not change the trends or conclusions of our work. Full implementation details are added to the Appendix D.
>
> ---
>
> > definitions of computer-use agent vs SWE agent and the rationale for “fair evaluation” (why expressiveness and access matter) could be better explained.
>
> Thank you for this feedback. We clarify the distinction as follows: a **computer-use agent (CUA)** perceives the screen as its observation and uses atomic keyboard and mouse actions to interact with the environment. In contrast, a specialized **SWE agent** is any agent developed specifically for software engineering tasks, typically using hand-engineered APIs. We have updated the paper to make this distinction clearer.
>
> The main goal of our work is to evaluate whether CUAs can perform SWE tasks. As a result, it is important that our environment is sufficiently **expressive**, i.e., allows for evaluating an agent on a wide range of tasks. In particular, we focus on the set of tasks achievable in an IDE, which is quite broad and to our knowledge could support all tasks that existing SWE agents are typically evaluated on.  Second, it’s important that the environment provides**access** to tools that specialized SWE agents might have access to. Otherwise, such SWE agents may have an unfair advantage. PwP is designed to satisfy both of these criteria. We agree that referring to these points as a “fair” comparison may be ambiguous, so we have removed this phrase and clarified the points above in the revision.
>
> ---
>
> > PwP-Bench-Lite covers many task types but has very few instances (20 per task), and the paper reports no repeated trials or variance, raising robustness concerns.
>
> Thank you for raising this point. To address robustness concerns, we have re-run the experiments for the best-performing computer-use agent (with and without bash/file APIs) over multiple trials. We find the variance in performance on PwP-Bench-Lite is very low (0.7%), and does not affect any of the conclusions of the work. Results are updated in Table 1 and 8.
>
> Further, the use of fewer instances per task is an intentional design choice to allow for computationally feasible evaluation across a wide diversity of SWE tasks.
>
> ---
>
> > Technically, the benchmark adapts existing datasets rather than introducing new ones.
>
>
> We agree that many tasks in PwP-Bench are adapted from existing datasets. We see this as a *key strength*, as it grounds our evaluation and allows for more direct comparison with the large body of work on specialized SWE agents.
>
> Furthermore, we do introduce two new datasets, *IDE Configuration* and *General-SWE*, which specifically test an agent's ability to interact with the IDE's tooling and complementing the existing benchmarks. Finally, although it is slightly tangential to your concern, we believe that creating an environment that is able to test agents on all of these datasets is itself a valuable contribution.
>
> ---
>
> Thank you again for your thoughtful review. We hope our responses and the additional experiment have addressed your concerns. If you think there are still some concerns, we look forward to addressing them as well! Alternatively, if you feel all your concerns are addressed, we would appreciate if you could reflect that in your evaluation!

---

> > ### Comment · Reviewer_f9o7 · 2025-11-26
> > **Thank you for you rebuttal**
> >
> > Thank you for carefully answering my questions. I wanted to clarify one of them.
> >
> > Regarding the comparison between the definitions of CUA and SWE agents, what I meant to ask is: what exactly are you comparing—the model’s input information, workflow, or the agents’ action space?
> >
> > I’m not closely following the literature on CUA and SWE agents, but I feel that providing a clearer breakdown of the differences between the two would help broaden the impact of the paper.

---

> > > ### Author Response · Authors · 2025-12-02
> > >
> > > Thank you for clarifying your question. We provide a concrete breakdown below:
> > >
> > > *Computer-Use Agents:*
> > >
> > > 1. Observation Space: Screen representation (e.g., screenshots or set-of-marks annotations)
> > > 2. Action Space: Keyboard and mouse interactions
> > > 3. Workflow: Sequence of alternating ⟨observation, action⟩ pairs
> > >
> > > *SWE Agents:*
> > >
> > > 1. Observation Space: Tool API outputs
> > > 2. Action Space: Tool API calls (e.g., file read, bash execute)
> > > 3. Workflow: Dependent on the agent designer. E.g., specific workflows in Agentless (Xia et al., 2024), or a more generic workflow as in mini-swe-agent (SWE-agent, 2024)
> > >
> > > We hope this breakdown and our previous response clarify the differences between the two types of agents! We have included this in the revised paper.

---

### Author Response · Authors · 2025-12-02

We thank all reviewers for their review of our work. We are encouraged that they find our work "addresses an important intersection between two major agent paradigms" with "novel environment design" and "insightful analysis" (Reviewer f9o7), that our "benchmark is very valuable" with "comprehensive" evaluation and ablation studies (Reviewer xKNb), that "PwP environment addresses a critical gap" with "extensible design" positioning it as "a lasting contribution" (Reviewer pzk4), that "evaluating CUAs on more complex tasks...is an important and valuable direction" with a "large and diverse benchmark" (Reviewer 5ZwW), that PwP is "the first comprehensive IDE-based environment" providing "clear empirical insights" and establishing "software engineering as a meaningful and economically significant domain" (Reviewer Syoz), and that our benchmark is "very challenging while staying close to real-world use cases," "solidly executed," and "well written and easy to understand" (Reviewer KWZS).

During the discussion period, we addressed the points raised by the reviewers and included several new experiments and clarifications in the revised paper: a) added OpenHands as additional SWE agent baseline (Reviewer f9o7), b) re-ran experiments over multiple trials showing low variance (Reviewers f9o7, pzk4), c) added quantitative IDE feature usage analysis in Figure 12, Appendix F (Reviewer xKNb), d) ran text-only mini-swe-agent baseline demonstrating importance of visual modality (Reviewer KWZS), e) added MoE model Qwen3-VL-30B-A3B (Reviewer pzk4), f) included all prompts in Appendix I and full task list in Appendix B (Reviewers KWZS, xKNb).

---

We want to highlight a few points about the discussion:
1. During the discussion, Reviewers xKNb and 5ZwW increased their scores (6 -> 8, 2 -> 4 respectively).
2. While Reviewer Syoz with score 4 did not respond, we believe that our response addresses their core concerns about cost/scalability (Appendix H), multi-step workflows (current agents struggle even on basic tasks), and evaluation metrics (we provide trajectory-level analysis beyond accuracy in Section 5.1); please see the discussion thread.
3. Reviewer 5ZwW left a final score of 4, but concluded that "our main disagreement is not about the technical soundness" and stated they "will not argue that it must be rejected." Please see the discussion thread for additional context.
4. The remaining reviewers ended with quite positive assessments of the paper.

---

We also affirm that we do not know who our reviewers are and had no interaction with them outside the system.

Finally, we want to thank the area chairs for making the best of a difficult situation.

---

### Meta-Review · Area_Chair_YYu7 · 2025-12-20

**Summary:**

The most prominent concern with this work noted by multiple reviewers was that it only compared to one SWE text only agent, Mini-SWE-agent. However, additional comparisons were then provided by the authors. There was also some concern about the lack of repeated trials and variances. A deeper analysis of debugger and profile usages (or not) was requested and the authors then added this to the appendix and it indicated that almost no debugging and profiler usage. Some concern was expressed that IDE interaction was not essential as text could be sufficient, but authors argue against this. This underlying disagreement persisted with the only reviewer giving the paper a reject rating.

**Reviewer Concerns:**

The key concern about not comparing with another text based SWE agent was addressed by adding comparisons to OpenHands. The concern about not enough replicates was addressed by adding such replicates. The concerns about the GUI not be that useful were somewhat addressed by adding experiments that explicitly focused on the utility of the visual information.

**Reviewer Scores:**

The most negative reviewer who had the philosophical position that the IDE GUI setting was not necessary and that text was sufficient did claim to have increased their score from 2 -> 4, ie still a reject rating, but they increased their score. Another reviewer was reported by the authors to have increased their score from 6->8 and this appears to be consistent to what was expressed in the available dialogues.

---

### Decision · Program_Chairs · 2026-01-26

Accept (Poster)